# Synergistic integration of Netrin and ephrin axon guidance signals by spinal motor neurons

**Sebastian Poliak[1,2†], Daniel Morales[3,4†], Louis-Philippe Croteau[3], Dayana Krawchuk[3,5], Elena Palmesino[3], Susan Morton[1,2,6], Jean-François Cloutier[4,7], Frederic Charron[3,8,9,10,11], Matthew B Dalva[12], Susan L Ackerman[5,13], Tzu-Jen Kao[3,14,15*], Artur Kania[3,8,4,9,10,11*]**

[1]Department of Neuroscience, Columbia University, New York, United States; [2]Department of Biochemistry and Molecular Biophysics, Columbia University, New York, United States; [3]Institut de recherches cliniques de Montréal, Montréal, Canada; [4]Integrated Program in Neuroscience, McGill University, Montréal, Canada; [5]The Jackson Laboratory, Bar Harbor, United States; [6]Howard Hughes Medical Institute, Columbia University, New York, United States; [7]Montréal Neurological Institute, Montréal, Canada; [8]Faculté de Médecine, Université de Montréal, Montréal, Canada; [9]Department of Biology, McGill University, Montréal, Canada; [10]Department of Anatomy and Cell Biology, McGill University, Montréal, Canada; [11]Division of Experimental Medicine, McGill University, Montréal, Canada; [12]Department of Neuroscience, The Farber Institute for Neurosciences, Jefferson Hospital for Neuroscience, Thomas Jefferson University, Philadelphia, United States; [13]Howard Hughes Medical Institute, The Jackson Laboratory, Bar Harbor, United States; [14]Graduate Institute of Neural Regenerative Medicine, College of Medical Science and Technology, Taipei Medical University, Taipei, Taiwan; [15]Center for Neurotrauma and Neuroregeneration, Taipei Medical University, Taipei, Taiwan

**\*For correspondence:** geokao@tmu.edu.tw (TJK); artur.kania@ircm.qc.ca (AK)

[†]These authors contributed equally to this work

**Competing interests:** The authors declare that no competing interests exist.

**Abstract** During neural circuit assembly, axonal growth cones are exposed to multiple guidance signals at trajectory choice points. While axonal responses to individual guidance cues have been extensively studied, less is known about responses to combination of signals and underlying molecular mechanisms. Here, we studied the convergence of signals directing trajectory selection of spinal motor axons entering the limb. We first demonstrate that Netrin-1 attracts and repels distinct motor axon populations, according to their expression of Netrin receptors. Quantitative in vitro assays demonstrate that motor axons synergistically integrate both attractive or repulsive Netrin-1 signals together with repulsive ephrin signals. Our investigations of the mechanism of ephrin-B2 and Netrin-1 integration demonstrate that the Netrin receptor Unc5c and the ephrin receptor EphB2 can form a complex in a ligand-dependent manner and that Netrin–ephrin synergistic growth cones responses involve the potentiation of Src family kinase signaling, a common effector of both pathways.

## Introduction

Axonal growth cones are guided to their targets by molecular cues laid down at discrete decision steps along their routes. This information is transduced inside growth cones via dedicated

**eLife digest** The ability of animals to walk and perform skilled movements depends on particular groups of muscles contracting in a coordinated manner. Muscles are activated by nerve cells called motor neurons found in the spinal cord. The connections between the motor neurons and muscles are established in the developing embryo. Each motor neuron produces a long projection called an axon whose growth is guided towards the target muscle by signal proteins. The motor neurons are exposed to many such signal proteins at the same time and it is not clear how they integrate all this information so that their axons target the correct muscles.

Poliak, Morales et al. used a variety of genetic and biochemical approaches to study the formation of motor neuron and muscle connections in the limbs of mice and chicks. The experiments show that a signal protein called Netrin-1 is produced in the limbs of developing embryos and attracts the axons of some types of motor neurons and repels others. This is due to the motor neurons producing different types of receptor proteins to detect Netrin-1.

Further experiments show that individual axons can combine information from attractive or repulsive Netrin-1 signals together with repulsive signals from another family of proteins called ephrins in a 'synergistic' manner. That is, the combined effect of both cues is stronger than their individual effects added together. This synergy involves ligand-dependent interactions between the Netrin-1 and ephrin receptor proteins, and the activation of a common enzyme.

Poliak, Morales et al.'s findings reveal a new role for Netrin-1 in guiding the development of motor neurons in the limb. Future work will focus on further understanding the mechanism of synergy between Netrin-1 and ephrins. Netrin-1 and ephrins are also involved in the formation of blood vessels and many other developmental processes, so understanding how they work together would have a wide-reaching impact on research into human health and disease.

intracellular signaling pathways, eventually modulating cytoskeletal dynamics. How multiple pathways interact and compound the limited molecular diversity of axon guidance cues to produce the rich complexity of nerve trajectories is unclear (*Dickson, 2002*; *Kolodkin and Tessier-Lavigne, 2011*; *Tessier-Lavigne and Goodman, 1996*).

Multiple modes of axon guidance signaling pathway interaction have been documented. Axon guidance ligand co-incidence can result in emergent growth cone behaviors that are qualitatively different from those induced by either cue alone (*Bielle et al., 2011*; *Kuwajima et al., 2012*). The exposure to one cue can also silence the response to another, as Slit silences Netrin-1 attraction in commissural axons, allowing them to continue their journey past the nervous system midline (*Stein, 2001*). Contrasting these is signal co-operation, where co-activation of two pathways changes the magnitude of growth cone responses, but not their quality. Co-operative steering of a growth cone in the same direction could entail two co-localized congruent cues, either both repellent or attractants, or two opposing cues, where a repellent reinforces the decision to grow towards an attractant cue (*Dudanova et al., 2010*; *Schmitt et al., 2006*). At the molecular level, co-operating axon guidance signaling pathways can intersect in additive or synergistic manners. Additive integration involves the summation of responses to individual cues implying parallel signaling pathways intersecting at the level of actin stability, as observed in additive phenotypes of mutations affecting motor axon guidance and muscle target selection in *Drosophila* and in vertebrates (*Kramer et al., 2006*; *Winberg et al., 1998*). Synergistic integration, that is, whose magnitude, when the signals are coincident, is greater than the sum of the effects of separate signals, implies signaling pathways intersecting at an intermediate step. It has been observed for congruent cues, such as bone morphogenetic proteins, whose repulsive signal integration occurs at the level of their receptors (*Butler and Dodd, 2003*; *Yamauchi et al., 2008*), and for Sonic Hedgehog and Ntn1 (*Sloan et al., 2015*). Despite this growing list of signal intersection modalities (*Dudanova and Klein, 2013*), their precise molecular mechanisms at a simple axon guidance waypoint are still unclear. One such point is traversed by spinal lateral motor column (LMC) axons as they enter the vertebrate limb where lateral and medial LMC axons diverge to form, respectively, dorsal and ventral limb nerves (*Lance-Jones and Landmesser, 1981*; *Tosney and Landmesser, 1985*). A molecular model of this event

implicates limb mesenchyme ephrin signaling to LMC growth cone tyrosine kinase Eph receptors. Lateral LMC growth cone expression of EphA receptors and medial LMC growth cone expression of EphB receptors results in their repulsion from cognate ephrin-A and ephrin-B ligands present in the ventral and dorsal limb mesenchyme, respectively (*Eberhart et al., 2000*; *Helmbacher et al., 2000*; *Kania and Jessell, 2003*; *Luria et al., 2008*). Additional mechanisms, such as reverse signaling from Eph receptors or semaphorins and glial cell-derived neurotrophic factor (GDNF) in the limb, co-operate with forward ephrin-A:EphA signaling to contribute to the fidelity of LMC axon trajectory choice (*Bonanomi et al., 2012*; *Chai et al., 2014*; *Dudanova et al., 2010*; *Huber et al., 2005*; *Kao and Kania, 2011*; *Kramer et al., 2006*; *Marquardt et al., 2005*). These observations imply a fragmented molecular logic of spinal motor axon guidance – ephrin:Eph forward signaling as a common effector of all LMC axon guidance, integrating with molecularly diverse LMC-subpopulation specific cues.

Netrins and their receptors have been implicated in diverse axon guidance events including motor axon exit from the spinal cord (*Bai et al., 2011*; *Hedgecock et al., 1990*; *Keino-Masu et al., 1996*; *Keleman and Dickson, 2001*; *Kennedy et al., 1994*; *Lai Wing Sun et al., 2011*; *Serafini et al., 1994*). The cellular response to Netrins is dictated by its receptors: deleted in colorectal cancer (DCC) and perhaps Neogenin enable attraction to Netrin-1 (*Keino-Masu et al., 1996*; *Palmesino et al., 2012*; *Xu et al., 2014*), while Unc5 proteins, in some cases in conjunction with DCC, elicit repulsion from Netrin-1 (*Hong et al., 1999*; *Leonardo et al., 1997*). Modulatory signals such as Slit and transforming growth factor β (TGF-β) act at the level of DCC to change Netrin-1 responses (*Bai et al., 2011*; *Leyva-Diaz et al., 2014*; *Stein, 2001*) or to set the sensitivity of repulsive Netrin receptors (*MacNeil et al., 2009*). The extensive modulation of Netrin-1 signaling raises the question of its relationship with ephrin signaling that is prominently involved in axon guidance decisions in the developing nervous system.

In this study, we provide genetic in vivo and in vitro evidence that limb mesenchyme Netrin-1 is a bifunctional effector, attracting lateral LMC and repelling medial LMC axons. Using in vitro growth preference assays, we demonstrate synergistic integration of congruent and opposing Netrin and ephrin signals. We also show that repulsive Netrin and ephrin receptors co-localize in a complex whose formation is modulated by ephrin-B2 signaling and EphB2 kinase activity, and that Src family kinase is a node for integration of Netrin-ephrin synergistic growth cone responses.

## Results

### Expression of Netrin-1 and its receptors in limb mesenchyme and LMC neurons

Netrin-1 (*Ntn1*) mRNA expression in developing limb buds (*Krawchuk and Kania, 2008*), prompted us to analyze the expression of Netrin ligands and their receptors at the time of LMC axon growth into the limb mesenchyme, between the embryonic day (e) 10.5 and e11.5 in mouse and between Hamburger-Hamilton stage (HH st.) 25 and 27 in chick (*Hamburger and Hamilton, 1951*; *Kania et al., 2000*; *Tosney and Landmesser, 1985*). Comparing Netrin family ligand mRNA expression with that of *Lmx1b*, a dorsal limb marker (*Riddle et al., 1995*), revealed that at the base of the mouse forelimb and hindlimb, where LMC axons select a dorsal or ventral limb trajectory, Netrin-1 mRNA and protein were enriched in the dorsal limb mesenchyme (*Figure 1A–D*). Expression of other Netrin ligands was either absent or not localized in a manner consistent with a role in LMC axon guidance at this choice point (*Figure 1—figure supplement 1*). In addition, the expression of β-galactosidase from an *Ntn1* gene trap allele (*Ntn1*[Gt]) recapitulated Netrin-1 mRNA and protein distribution (*Figure 1E,F*; *Serafini et al., 1996*). In HH st. 25 chickens, *Ntn1* mRNA was also confined to the dorsal limb (*Figure 1G,H*).

To determine whether Netrin-1 receptors are present in developing LMC neurons, we first compared their mRNA expression in mouse spinal cord sections with that of medial and lateral LMC neuron markers, *Isl1* and *Lhx1*, respectively, between e10.5 and e11.5 (*Tsuchida et al., 1994*). *Unc5c*, a repulsive Netrin-1 receptor gene, was expressed in Isl1[+] medial LMC neurons (*Figure 1I,J*), while *Dcc* and Neogenin1 (*Neo1*), encoding attractive Netrin-1 receptors, were expressed by both LMC divisions at hindlimb and forelimb levels (*Figure 1K,L*; *Figure 1—figure supplement 1*). Detection of Unc5c protein using an antibody raised against its C terminus, and Isl1 and Lhx1, confirmed that Unc5c is present in medial LMC neurons, but not in lateral LMC neurons (*Figure 1—figure*

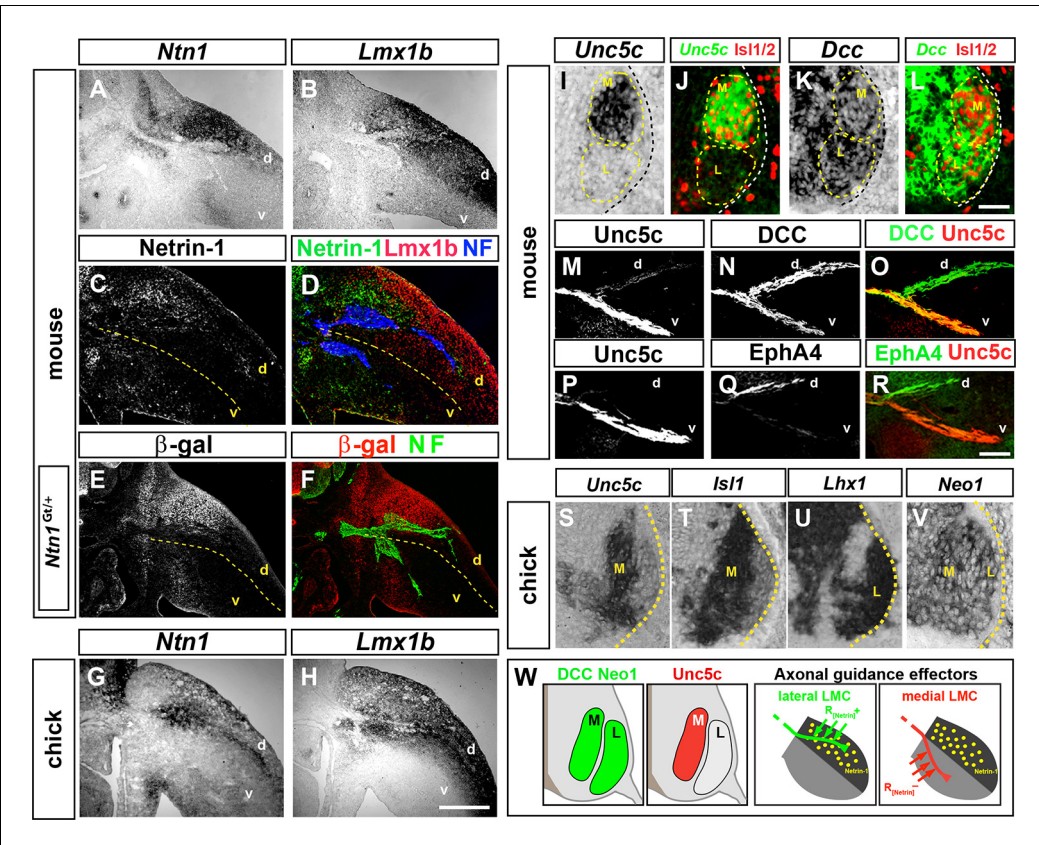

**Figure 1.** Expression of Netrin-1 in the limb mesenchyme and of Netrin receptors in medial and lateral LMC neurons. Netrin-1 mRNA and protein detected by in situ hybridization and immunohistochemistry compared with the dorsal limb marker Lmx1b in mouse and chick limbs at the time of LMC axon limb ingrowth. (**A, B**) In situ hybridization for *Ntn1* (**A**) and the dorsal limb marker *Lmx1b* (**B**) mRNAs in consecutive sections of e11.5 wild-type mouse forelimb. (**C, D**) Immunostaining for Netrin-1, neurofilament, and Lmx1b in e11.5 mouse forelimb. (**E, F**) Immunostaining for β-galactosidase and neurofilament in e11.5 $Ntn1^{Gt/+}$ mouse forelimb. β-galactosidase is expressed at the dorsal limb mesenchyme domain abutting dorsally projecting LMC axons. (**G, H**) In situ hybridization for *Ntn1* mRNA (**G**) and *Lmx1b* mRNA (**H**) in consecutive sections of HH st. 27 chick hindlimbs. *Unc5c* and *DCC* (or *Neogenin* in chick) mRNA and protein expression was compared with LMC divisional markers in e11.5 mice and HH st. 27 chick spinal cord or to EphA4 in lateral LMC axons. (**I, J**) Selective expression of Unc5c in medial LMC motor neurons. In situ detection of *Unc5c* mRNA and immunodetection of Isl1/2 in e11.5 lumbar mouse spinal cord. The green signal in (**J**) represents the pseudocolor image shown in (**I**). M and L indicate the position of medial and lateral LMC, respectively, as assessed by Isl1/2 (red) or Lhx1 immunostaining (not shown). (**K, L**) DCC is expressed at both LMC divisions. In situ hybridization of *Dcc* mRNA and immunodetection of Isl1/2 in e11.5 mouse lumbar spinal cord. The green signal represents the pseudocolor image. Note the higher level of *Dcc* in lateral (Isl1[-]) vs. medial (Isl1[+]) LMC motor neurons. (**M–R**) Localization of Unc5c and DCC proteins in dorsal and ventral axon branches entering the hindlimb in e11.5 embryos. (**M–O**) Double immunostaining for Unc5c and DCC. Note the high expression level of Unc5c in ventral nerves and DCC in both dorsal and ventral nerves. (**P–R**) Double immunostaining for Unc5c and EphA4. (**S–V**) Expression of Netrin-1 receptors in chick LMC neurons. In situ hybridization for *Unc5c, Neo1, Isl1*, and *Lhx1* in consecutive sections of HH St. 27 chick lumbar spinal cord. Note the presence of in medial (*Isl1[+]*) LMC neurons and of *Neo1* in both LMC divisions. (**W**) Summary of Netrin-1, DCC, and Unc5c expression in limb mesenchyme and LMC neurons. LMC, lateral motor column; Ntn1, Netrin-1; NF, neurofilament; M, medial; L, lateral; d, dorsal; v, ventral. Scale bars: (A–H) 250 µm, (I–L) 80 µm, (M–V) 50 µm.

The following figure supplement is available for figure 1:

**Figure supplement 1.** Expression of Netrin family ligands and receptors in mouse and chick hindlimbs.

supplement 1; data not shown). Co-detection of DCC and Isl1 or Lhx1 proteins indicated that DCC is expressed by both LMC divisions (*Figure 1—figure supplement 1*; data not shown). Examination of motor and sensory nerves near their spinal cord and dorsal root ganglion exit, before they merge into mixed sensory-motor nerves, revealed that DCC is present exclusively on motor axons, while Unc5c is expressed at high levels in motor axons with some expression in sensory axons (*Figure 1—figure supplement 1*). In the limb, high levels of Unc5c protein were present in ventral limb nerves (*Figure 1M–O*), consistent with its medial LMC expression, while DCC was detected in both dorsal and ventral nerves (*Figure 1N–O*), consistent with its pan-LMC expression. Additional analysis revealed that mouse *Dscam* and *Unc5a* were expressed in all LMC neurons, while *Unc5b* and *Unc5d* were not detected in this population (*Figure 1—figure supplement 1*). Chick *Unc5b* and *Unc5c* were expressed in medial LMC neurons while *Unc5b* and *Neo1* mRNAs were found in all LMC neurons (*Figure 1S–V*, *Figure 1—figure supplement 1*). Altogether, these studies suggested that dorsal limb Ntn1 may play a dual role in LMC axon guidance, attracting lateral LMC axons that do not express Unc5c into the dorsal limb, and repelling Unc5c-expressing medial LMC axons into the ventral limb (*Figure 1W*).

## Netrin signaling is required for LMC axon guidance in the limb

We next determined whether loss of Netrin-1 or its receptors affects LMC axon pathfinding, by analyzing the limb trajectories of medial and lateral LMC axons in e11.5 and e12.5 wild-type, $Ntn1^{Gt/Gt}$, $Dcc^{-/-}$, $Neo1^{Gt/Gt}$, $Unc5c^{-/-}$, and $Unc5a^{-/-}$ mutant mice (*Bae et al., 2009*; *Boyer and Kozak, 1991*; *Burgess et al., 2006*; *Fazeli et al., 1997*; *Williams et al., 2006*). In all strains examined, the specification of LMC neurons, their axon outgrowth, and Eph and ephrin expression was normal (*Figure 2—figure supplement 2*; data not shown). To examine LMC axon trajectory selection, we injected the retrograde tracer horseradish peroxidase (HRP) into the ventral or dorsal limb muscles of $Ntn1^{Gt/Gt}$ and Netrin receptor mutant embryos and determined the divisional identity of labeled LMC neurons using the marker Foxp1 (*Dasen et al., 2008*; *Rousso et al., 2008*), and the medial LMC marker Isl1 (*Luria et al., 2008*). All our retrograde tracing observations were confirmed using an alkaline phosphatase medial LMC reporter transgene or Unc5c and EphA4 axonal staining (*Figure 2—figure supplement 1*). When HRP was injected into the ventral limb to detect aberrant lateral LMC axons, in *Dcc* mutants 9.7% of HRP-labeled neurons were medial LMC, compared with 4.4% in controls, 4.7% in $Ntn1^{Gt/Gt}$ and 5.6% in $Unc5c^{-/-}$ mice (*Figure 2A–H*; p=0.164, 1, and 0.748 for $Dcc^{-/-}$, $Ntn1^{Gt/Gt}$, and $Unc5c^{-/-}$ vs. controls; all values listed in *Supplementary file 1B*). We also injected HRP into the ventral limbs of $Neo1^{Gt/Gt}$ or $Dscam^{-/-}$ single or $Dcc^{-/-}$; $Neo1^{Gt/Gt}$ double mutants and assessed the divisional identity of labeled LMC neurons (*Figure 2—figure supplement 2*). Neither single nor double mutant lateral LMC axons were found in the ventral limb at an incidence greater than that in control embryos (*Figure 2—figure supplement 2*; p=0.105, 0.537 and 0.537 for $Neo1^{Gt/Gt}$, $Dscam^{-/-}$ and $Dcc^{-/-}$; $Neo1^{Gt/Gt}$ vs. wild-type controls). Thus, extensive redundancy notwithstanding, *Dcc*, Neogenin, or *Dscam* are not required for in vivo lateral LMC axon guidance.

Next, to detect aberrant medial LMC axon projections, we injected HRP into the dorsal limb of control and mutant mice and the proportion of labeled medial motor neurons determined as a percentage of all labeled motor neurons. In wild-type mice, 3.9% of tracer-labeled LMC neurons were medial LMC (*Figure 2K,L,S*), whereas in *Unc5c* mutants, 40.4% of labeled LMC neurons were medial LMC, corresponding to essentially random limb nerve selection (*Figure 2Q,R,S*; p<0.001 wild-type vs. $Unc5c^{-/-}$). In $Ntn1^{Gt/Gt}$ mice, 12.6% of labeled LMC neurons were medial LMC, representing a significant projection error (*Figure 2O,P,S*; p<0.05 vs. wild-type). Similar incidence of misprojections in $Ntn1^{Gt/Gt}$ and *Unc5c* mutants were also observed using a medial LMC-specific transgene (*Figure 2—figure supplement 1*), suggesting that Ntn1 in the dorsal limb repels medial LMC axons through Unc5c.

As Unc5c-DCC heterodimers have been proposed to mediate repulsion from Ntn1 (*Hong et al., 1999*), we asked whether DCC is required to specify the limb trajectory of medial LMC axons. In $Dcc^{-/-}$ mice with dorsal limb tracer injection, medial LMC neurons represented 3.3% of labeled LMC neurons, a proportion not significantly different from the 3.9% observed in controls (*Figure 2M,N,S*, p=1 vs. wild-type controls). To determine whether *Neo1* might act with DCC to mediate repulsion from Ntn1, we investigated the fidelity of medial LMC axon guidance in $Neo1^{Gt/Gt}$ single or $Dcc^{-/-}$; $Neo1^{Gt/Gt}$ double mutants. Medial LMC projections in $Neo1^{Gt/Gt}$ or in $Dcc^{-/-}$; $Neo1^{Gt/Gt}$ double mutants were not different from controls, and neither were they affected in $Dscam^{-/-}$ mice

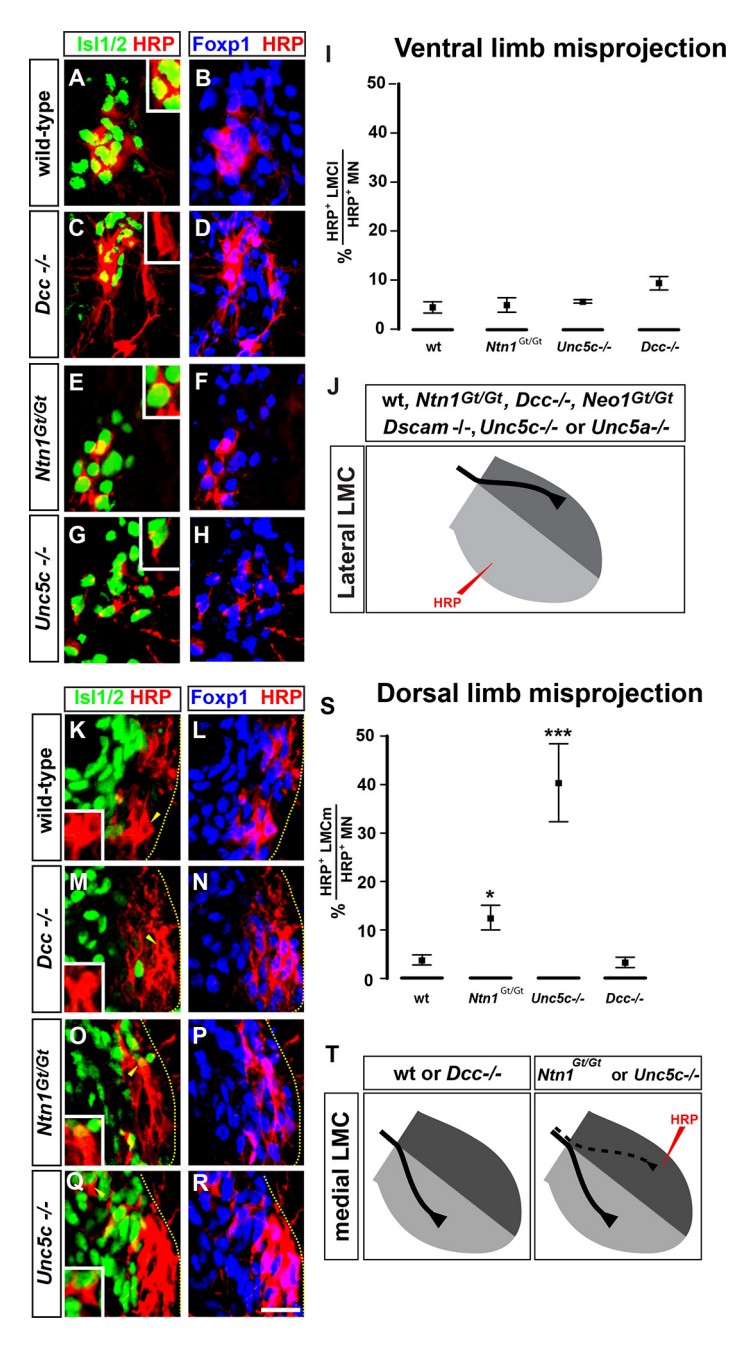

**Figure 2.** The requirement of Netrin-1 and its receptors for the fidelity of LMC axon trajectory selection. Lateral and medial LMC axon projections were analyzed by injecting the retrograde tracer HRP into dorsal or ventral limb muscles followed by the assessment of LMC divisional identity of backfilled LMC neurons. (A–H) Analysis of lateral LMC motor axon projections in wild-type (A and B), $Dcc^{-/-}$ (C and D), $Ntn1^{Gt/Gt}$ (E and F) and $Unc5c^{-/-}$ (G and H) mutant mice. A retrograde tracer (HRP, red) was injected in the ventral forelimb of e12.5 wild-type or mutant mice followed by detection of Isl1 (green, A, C, E, G) and Foxp1 (blue, B, D, F, H) to identify medial (Isl1$^+$, Foxp1$^+$) and lateral (Isl1$^-$, Foxp1$^+$) LMC neurons. Insets in A, C, E and G show examples of magnified HRP$^+$ backfilled cells that are Isl1$^-$ ($Dcc^{-/-}$ mice) or Isl1$^+$ (wild-type, $Ntn1^{Gt/Gt}$ or $Unc5c^{-/-}$ mice). (I) Quantification of retrogradely labeled lateral LMC axon projections. The graph depicts the mean percentage ± SD of HRP$^+$ backfilled motor neurons that express the medial LMC marker Isl1 after a dorsal limb injection. n = 5 (wild-type), 4 ($Ntn1^{Gt/Gt}$), 3 ($Unc5c^{-/-}$), 5 ($Dcc^{-/-}$) embryos. (J) Summary of analysis of lateral LMC projections in different Netrin signaling mutant mice. See *Figure 2—figure supplement 2* for results of additional mutant analyses. (K–R) Analysis of medial LMC motor axon projections in wild-type (K and L), $Dcc^{-/-}$ (M and N), $Ntn1^{Gt/Gt}$ (O and P), and $Unc5c^{-/-}$ (Q and R) mutant mice.
*Figure 2 continued on next page*

*Figure 2 continued*

HRP (red) was injected in the dorsal forelimb of e12.5 wild-type or mutant mice followed by immunostaining of spinal cord sections for Isl1 (green, **K, M, O, Q**) and Foxp1 (blue, **L, N, P, R**) to identify medial (Isl1$^+$, Foxp1$^+$) and lateral (Isl1$^-$, Foxp1$^+$) LMC neurons. Arrowheads indicate examples of HRP$^+$ cells that are Isl1$^-$ (wild-type and *Dcc*$^{-/-}$ mice) or Isl1$^+$ (*Ntn1*$^{Gt/Gt}$ or *Unc5c*$^{-/-}$ mice) and are magnified in the insets. (**S**) Quantification of retrogradely labeled medial LMC axon projections. The graph depicts the mean percentage ± SD of HRP$^+$ backfilled motor neurons that express the medial LMC marker Isl1 after a dorsal limb injection. n = 3 (wild-type), 3 (*Ntn1*$^{Gt/Gt}$), 5 (*Unc5c*$^{-/-}$), 3 (*Dcc*$^{-/-}$) embryos. (**T**) Summary scheme of medial LMC projections in wild-type, *Ntn1*$^{Gt/Gt}$, *Unc5c*$^{-/-}$, and *Dcc*$^{-/-}$ mice. In *Ntn1*$^{Gt/Gt}$ and *Unc5c*$^{-/-}$ mice some medial LMC axons project to the dorsal limb. HRP, horseradish peroxidase; wt, wild-type; Ntn1, Netrin-1; error bars = SD; *** = p<0.001; * = p<0.05; statistical significance computed using Fisher's exact test. All values (mean ± SD) can be found in ***Supplementary file 1B***; scale bar: 20 μm.

The following figure supplements are available for figure 2:

**Figure supplement 1.** Analysis of medial LMC axon projections in mutant mice.

**Figure supplement 2.** Normal specification of LMC neurons in *Unc5c*, *Dcc*, and *Ntn1*$^{Gt}$ mutants and summary of LMC axon trajectory analysis in mutant mice.

---

(***Figure 2—figure supplement 2***; p=0.537, 0.748, 0.537 for *Neo1*$^{Gt/Gt}$, for *Dcc*$^{-/-}$; *Neo1*$^{Gt/Gt}$, and for *Dscam*$^{-/-}$ vs. wild-type controls, respectively), suggesting that attractive Ntn1 receptors are not required for medial LMC axon guidance.

## Netrin-1 guides medial and lateral LMC axons in vitro

We next assessed whether Netrin-1 can directly modulate the behavior of LMC axons by monitoring the response of medial and lateral LMC axons to Netrin-1 protein in an in vitro stripe assay (***Kao and Kania, 2011***). We exposed chick LMC axons to protein carpets in alternating stripes of either Fc protein and Cy3-labeled secondary antibody mixed with Fc (Fc/Fc) or Fc protein and Cy3-labeled secondary antibody mixed with Netrin-1 (Fc/N). In all stripe assay experiments, the total amount of LMC axon outgrowth was similar (data not shown). Lateral LMC axons, identified by EphA4 expression, did not show significant stripe preference when grown on Fc/Fc stripes, with similar proportion of axons growing over either stripe (***Figure 3A***, 48% on Cy3-Fc vs. 52% on Fc). However, when confronted with Fc/N stripes, lateral LMC axons preferentially grew on Netrin-1 stripes (***Figure 3B***; 75% on N, 25% on Fc stripes p<0.001 vs. Fc/Fc controls).

In chick, the attractive Netrin receptor function has been proposed to be carried out by Neogenin, since there is no *Dcc* gene in this species (***Phan et al., 2011***). To address this possibility, we performed an Fc/N stripe preference assay incubating LMC axons in the presence of a function-blocking antibody raised against the extracellular domain of *Neo1*. In such experiments, LMC axon preference for Ntn1 stripes was abolished (***Figure 3C***, 53% on N, 47% on Fc stripes p<0.001 vs. untreated N/Fc; [***Palmesino et al., 2012***]), but could be rescued by electroporation of rat *Dcc* expression plasmid (***Figure 3E***; 66% on N, 34% on Fc stripes) but not with a plasmid encoding a truncated DCC (*Dcc*$^{\Delta ICD}$; ***Figure 3D***; 48% on N, 52% on Fc stripes; p<0.001 vs. *Dcc* overexpression). Lateral LMC axons incubated with anti-*Neo1* antibodies did not lose their sensitivity to ephrin-A5 (***Figure 3—figure supplement 1***). These experiments show that *Neo1* mediates lateral LMC neurite growth preference on Netrin-1, and that DCC is functionally equivalent.

To study the behavior of medial LMC axons in response to Netrin-1, we explanted green fluorescent protein (GFP)-expressing LMC neurons from chick embryos electroporated with the medial LMC-specific expression plasmid e[Isl1]::GFP (***Kao et al., 2009***). While GFP-expressing medial LMC axons did not favor either control stripe (***Figure 3F***; 51% on Cy3-Fc, 49% on Fc stripes), when challenged with Fc/Netrin-1 stripes, medial LMC axons avoided Netrin-1 in favor of Fc (***Figure 3G***; 24% on N, 76% on Fc stripes, p<0.001). This repulsive effect was abolished in medial LMC neurons electroporated with the siRNA targeting *Unc5c*, an effect that was rescued by expression of mouse Unc5c (***Figure 3H,I***; 48% on N, 52% on Fc stripes, p<0.01; 28% on N, 72% on Fc stripes, p=0.281). Inclusion of a function-blocking Neogenin antibody in the stripe assays did not block the repulsion of medial LMC neurites from Netrin-1 (***Figure 3L***; 27% on N, 73% on Fc stripes, p=0.113). Thus, Netrin-1 causes repulsion of medial LMC axons through Unc5c, independently of Neogenin,

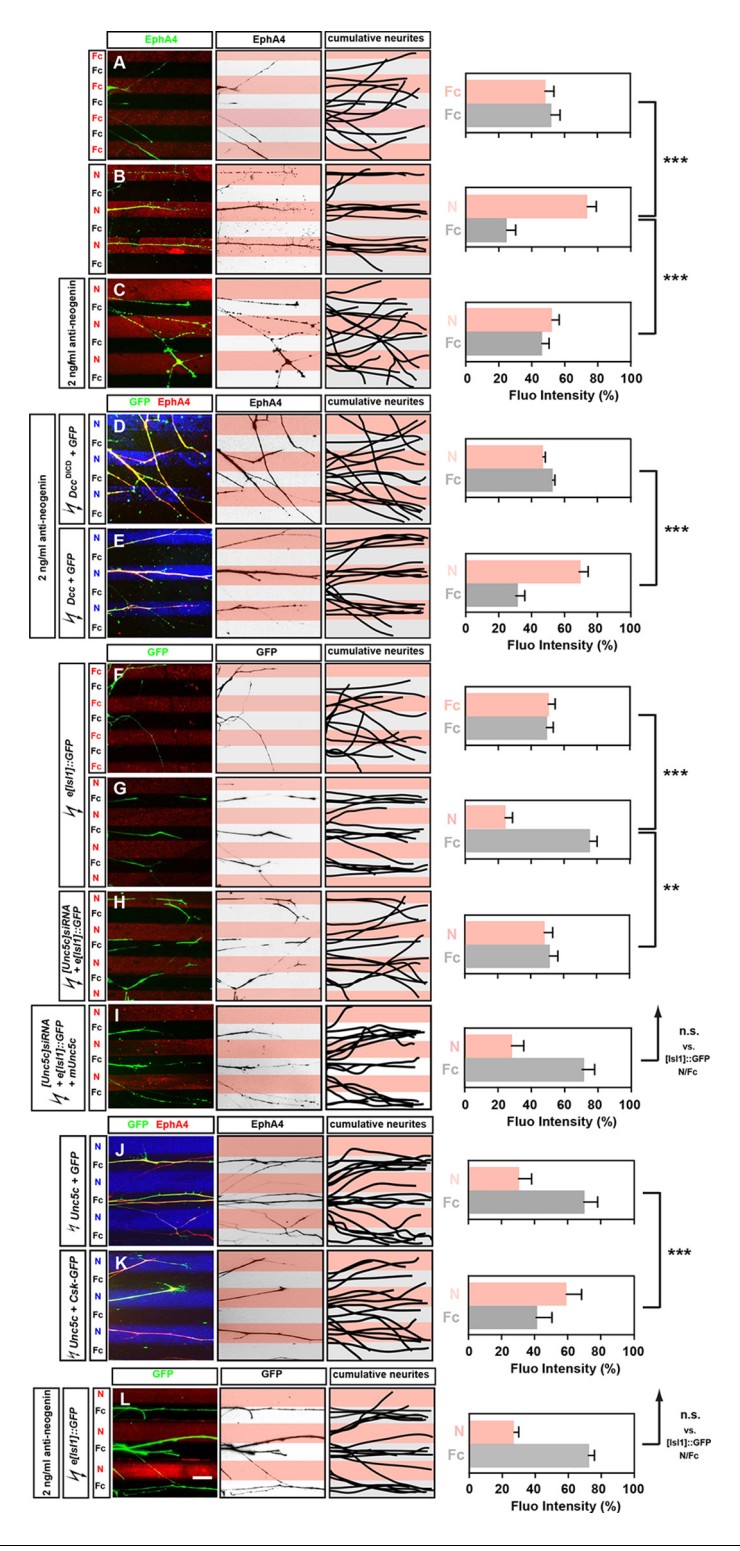

**Figure 3.** Opposing responses of medial and lateral LMC axons to Netrin-1 require Neogenin and Unc5c function. Growth preference on protein stripes exhibited by medial and lateral LMC axons. Each experiment is composed of three panels (left, middle, right) and a quantification. (A–E) Left panels: explanted lateral (EphA4⁺) LMC neurites on Fc/Fc (A) or Netrin-1 (N)/Fc stripes without (B) or with (C) the addition of anti-neogenin antibody. Lateral (GFP⁺ EphA4⁺) LMC neurites of $Dcc^{\Delta ICD}$ and $GFP$ (D) or $Dcc$ and $GFP$ (E) co-electroporated explants treated with anti-neogenin antibody. Middle panels: inverted images where EphA4 signal is dark pixels overlaid on substrate

*Figure 3 continued on next page*

*Figure 3 continued*

stripes. Right panels: superimposed images of five representative explants from each experimental group highlighting the distribution of lateral LMC neurites. Quantification of lateral (EphA4$^+$) LMC neurites on first (pink) and second (pale) stripes expressed as a percentage of total EphA4 signals. Number of neurites: 77. Minimal number of explants: 11. (F–I) Left panels: detection of medial (GFP$^+$) LMC neurites of explants on Fc/Fc (F) or N/Fc stripes (G), [Unc5c]siRNA co-electroporated explants on N/Fc stripes (H) and [Unc5c]siRNA + Unc5c co-electroporation rescue experiment (I). Middle panels: inverted images where GFP signal is dark pixels overlaid on substrate stripes. Right panels: superimposed images of five representative explants from each experimental group highlighting the distribution of medial LMC neurites (right panels). Quantification of medial (GFP$^+$) LMC neurites on first (pink) and second (pale) stripes expressed as a percentage of total GFP signals. Minimal number of neurites: 83. Minimal number of explants: 12. (J–K) Left panels: explanted lateral (EphA4$^+$) LMC neurites on N/Fc stripes. Lateral (GFP$^+$ EphA4$^+$) LMC neurites of Unc5c and GFP (J) or Unc5c and Csk-GFP (K) co-electroporated explants. Middle panels: inverted images where EphA4 signal is dark pixels overlaid on substrate stripes. Right panels: superimposed images of five representative explants from each experimental group highlighting the distribution of lateral LMC neurites. Quantification of lateral (EphA4$^+$) LMC neurites on first (pink) and second (pale) stripes expressed as a percentage of total EphA4 signals. Minimal number of neurites: 90. Minimal number of explants: 11. (L) Left panels: detection of medial (GFP$^+$) LMC neurites of explants on N/Fc stripes with anti-neogenin antibody addition. Middle panels: inverted images where GFP signal is dark pixels overlaid on substrate stripes. Right panels: superimposed images of five representative explants from each experimental group highlighting the distribution of medial LMC neurites (right panels). Quantification of medial (GFP$^+$) LMC neurites on first (pink) and second (pale) stripes expressed as a percentage of total GFP signals. Minimal number of neurites: 83. Minimal number of explants: 12. LMC, lateral motor column; N, netrin-1; error bars = SD; *** = p<0.001; ** = p<0.01; n.s. = not significant; statistical significance computed using Mann-Whitney U test; all values (mean ± SD) can be found in *Supplementary file 1B; s*cale bar: 50 µm.
The following figure supplement is available for figure 3:

**Figure supplement 1.** Functional blocking of Neogenin has no effect on ephrin-mediated lateral LMC repulsion, quantification of Unc5c-expressing medial LMC neurons in mice and chick.

providing direct evidence that Netrin-1 is a bifunctional axon guidance cue for LMC axons. Finally, we asked whether expression of Unc5c is sufficient to evoke LMC axon repulsion from Netrin-1. Explanted lateral LMC neurons electroporated with an Unc5c expression plasmid avoided Netrin-1 stripes. UNC5-mediated repulsion from Netrin-1 has been shown to depend on the Src kinase (*Williams et al., 2006*), and indeed the repulsion of Unc5c-expressing LMC axons from Netrin-1 stripes was abolished by the expression of the Src-inhibiting kinase Csk (*Figure 3J,K*; 30% on N, 70% on Fc stripes vs. 59% on N, 41% on Fc stripes, p<0.001; [*Imamoto and Soriano, 1993*]).

## In vivo co-operation of Unc5c and EphB2 in LMC axon guidance

Given the prominent function of ephrins in motor axon guidance, we next investigated the possibility that Netrin and ephrin signals co-operate in LMC neurons. To do this we focused on the function of Unc5c and EphB2, an ephrin-B receptor guiding medial LMC axons in vivo (*Luria et al., 2008*). We first tested whether chicken Unc5c is required in vivo by reducing its expression through *[Unc5c] siRNA* and *GFP* plasmid electroporation of presumptive LMC neurons at HH st. 17–19. Analysis of the proportion of electroporated LMC axons entering the dorsal versus ventral limb nerves at HH st. 26 showed that in embryos electroporated with *[Unc5c]siRNA*, 72% of GFP$^+$ axons were present in the dorsal limb nerve, compared with 52% in *GFP* electroporated controls (*Figure 4A,B*; *Figure 4— figure supplement 1*; p<0.01), demonstrating that chicken Unc5c is required for the fidelity of LMC trajectory choice.

To test whether Unc5c and EphB2 co-operate to direct LMC axons into the ventral limb, we compared the limb trajectory of LMC axons overexpressing GFP, Unc5c and GFP, EphB2 fused to GFP (EphB2-GFP) or Unc5c together with EphB2-GFP. On their own, Unc5c and EphB2-GFP caused a significantly greater fraction of GFP$^+$ axons to select the ventral limb nerve when compared with GFP-only controls (*Figure 4A,C,D*; ventral limb: 48% in GFP, 57% in *Unc5c/GFP*, p<0.05 vs. GFP; 64% in Ephb2-GFP, p<0.01 vs. GFP). However, when co-expressed, Unc5c and EphB2-GFP caused significantly more GFP$^+$ axons to select the ventral limb nerve branch (*Figure 4E*; ventral branch: 80% in

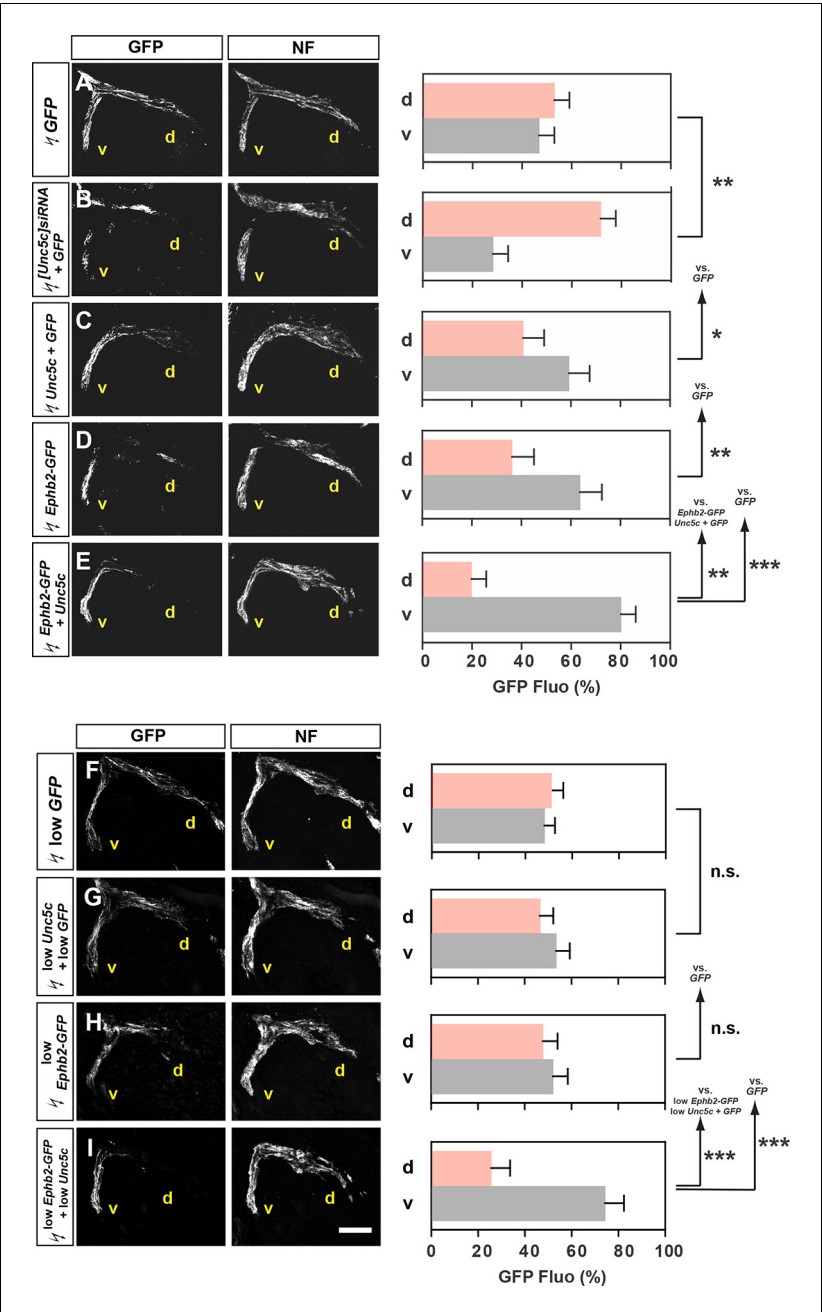

**Figure 4.** Co-operation between Unc5c and EphB2 receptors in LMC trajectory selection. (**A–E**) GFP and neurofilament detection in the limb nerve branches in the crural plexus of embryos electroporated with the following expression plasmids and siRNAs: *GFP* (**A**), *[Unc5c]siRNA* and *GFP* (**B**), *Unc5c* and *GFP* (**C**), *EphB2-GFP* (**D**), or *EphB2-GFP* and *Unc5c* (**E**). Quantification of GFP signals in all electroporation experiments expressed as, respectively, percentage in dorsal and ventral limb nerves (GFP Fluo [%]). n = 5 embryos. (**F–I**) GFP and neurofilament detection in the limb nerve branches in the crural plexus of embryos electroporated with the following expression plasmids with 20% of normal concentration: low *GFP* (**F**), low *Unc5c* and low *GFP* (**G**), low *EphB2-GFP* (**H**), or low *EphB2-GFP* and low *Unc5c* (**I**). Quantification of GFP signals in all electroporation experiments expressed as, respectively, percentage in dorsal and ventral limb nerves (GFP Fluo [%]). n = 5 embryos. See text for detailed description. d, dorsal; v, ventral; error bars = SD; *** = p<0.001; ** = p<0.01; * = p<0.05; n.s.= non significant; statistical significance computed using Mann-Whitney U test; All values (mean ± SD) can be found in *Supplementary file 1B*. Scale bar: 150 µm.

The following figure supplement is available for figure 4:

*Figure 4 continued on next page*

*Figure 4 continued*

**Figure supplement 1.** Characterization of siRNA-mediated Unc5c knockdown and Unc5c overexpression in LMC neurons in chicken.

*Unc5c* and *Ephb2-GFP*, p<0.01 vs. *Unc5c/GFP*; p<0.01 vs. *Ephb2-GFP*), suggesting that EphB2 and Unc5c can co-operate to guide LMC axon guidance.

We next assessed whether electroporation of combined Unc5c and EphB2 at low concentrations elicits an effect not observed when each receptor expression plasmid is electroporated alone. To do this, we electroporated low concentrations of *Ephb2-GFP, Unc5c*, and *GFP*, or *Ephb2-GFP* and *Unc5c* plasmids into LMC neurons, and compared spinal cord section Unc5c, GFP, and EphB2 protein and mRNA levels against standard DNA level electroporations. We estimated that EphB2 and Unc5c proteins produced from the plasmids electroporated at low concentrations were approximately 30% of that induced by electroporating high plasmid concentrations (*Figure 4—figure supplement 1*). The proportion of LMC axons entering the ventral limb nerve in such *Unc5c/GFP* or *Ephb-GFP* embryos was not significantly different from controls (*Figure 4F–H*; both p>0.107). However, LMC axons co-expressing low levels of *Ephb2-GFP* and *Unc5c* entered the ventral limb nerve with an incidence of 74%, a proportion significantly different from controls and confirmed by retrograde tracer injection into the ventral limb (*Figure 4I*; p<0.001; *Figure 4—figure supplement 1*). Together, these results suggest in vivo LMC co-operative responses to ephrin-B and Ntn1 could be synergistic in nature.

## Medial and lateral LMC axons synergistically integrate congruent and opposing Netrin and ephrin signals

To directly determine how ephrin and Netrin-1 signals co-operate in LMC axons, we took advantage of *e[Isl1]::GFP*-electroporated medial LMC axons' preferential growth on Fc stripes when confronted with Fc/ephrin-B2 (eB2) stripes, generated by incubation with a standard eB2 concentration ($C_{100}$ eB2=10 µg/ml; *Figure 5A*; 79% on Fc, 22% on eB2; [*Kao and Kania, 2011*]). This preference was significantly increased when medial LMC neurites were confronted with Fc/eB2+Netrin-1 (each at $C_{100}$) stripes mimicking the restriction of eB2 and Netrin-1 to the dorsal limb mesenchyme (*Figure 5B*; 90% on Fc, 10% on eB2+Netrin-1, p<0.05 vs. eB2; $C_{100}$ Netrin-1 = 100 ng/ml). The increased LMC axon repulsion from ephrins in the presence of Netrin-1 could be due to Netrin and Eph signaling acting simultaneously at the level of single growth cones, or due to recruitment of non-overlapping subpopulations of LMC neurons that only express one of the receptor families. The vast majority of LMC neurons co-express both types of receptors (*Kania and Jessell, 2003*; *Luria et al., 2008*; *Figure 1I–R*; *Figure 3—figure supplement 1*; data not shown), arguing that ephrins and Ntn1 can co-operatively act in individual LMC axons.

To address whether Netrin-1-ephrin co-operation is additive or synergistic in nature we performed ligand titration experiments. We considered the possibility that simultaneous application of Netrin-1 and ephrin-B2 at concentrations that do not elicit a stripe preference on their own could result in their summation exceeding a threshold required for a preference response. Thus, signals at subthreshold concentrations could elicit a growth cone response when presented together, through simple additive means. However, we reasoned that a stripe preference evoked by simultaneous exposure to Netrin-1 and eB2 concentrations that are each *half or less* of those subthreshold concentrations not eliciting stripe preference when alone, would indicate synergistic co-operation between Netrin and ephrin signaling since simple arithmetic additivity could not account for the crossing of a threshold. Thus, we tested whether a reduction to 50%, 25%, and 10% of our standard ephrin-B2 and Netrin-1 concentrations ($C_{50}$, $C_{25}$, and $C_{10}$, respectively) is sufficient to abolish the stripe assay response by LMC growth cones to individual cues, and whether, at these concentrations, their coincidence rescues stripe preference. Medial LMC neurons challenged with Fc stripes versus stripes with Netrin-1 at $C_{50}$, $C_{25}$, or $C_{10}$ displayed no preference (*Figure 5C*; *Figure 5—figure supplement 1*; at $C_{10}$ 48% neurites on Netrin-1; p=0.081), whereas ephrin-B2 exposure resulted in reduced preference at $C_{50}$, and no detectable preference at $C_{25}$, and $C_{10}$ (*Figure 5D*; *Figure 5—figure supplement 1*; at $C_{10}$ 45% neurites on eB2; p=0.081). However, when Netrin-1 and eB2 were presented together in the same stripes, alternating with Fc stripes, we observed a striking repulsion of medial

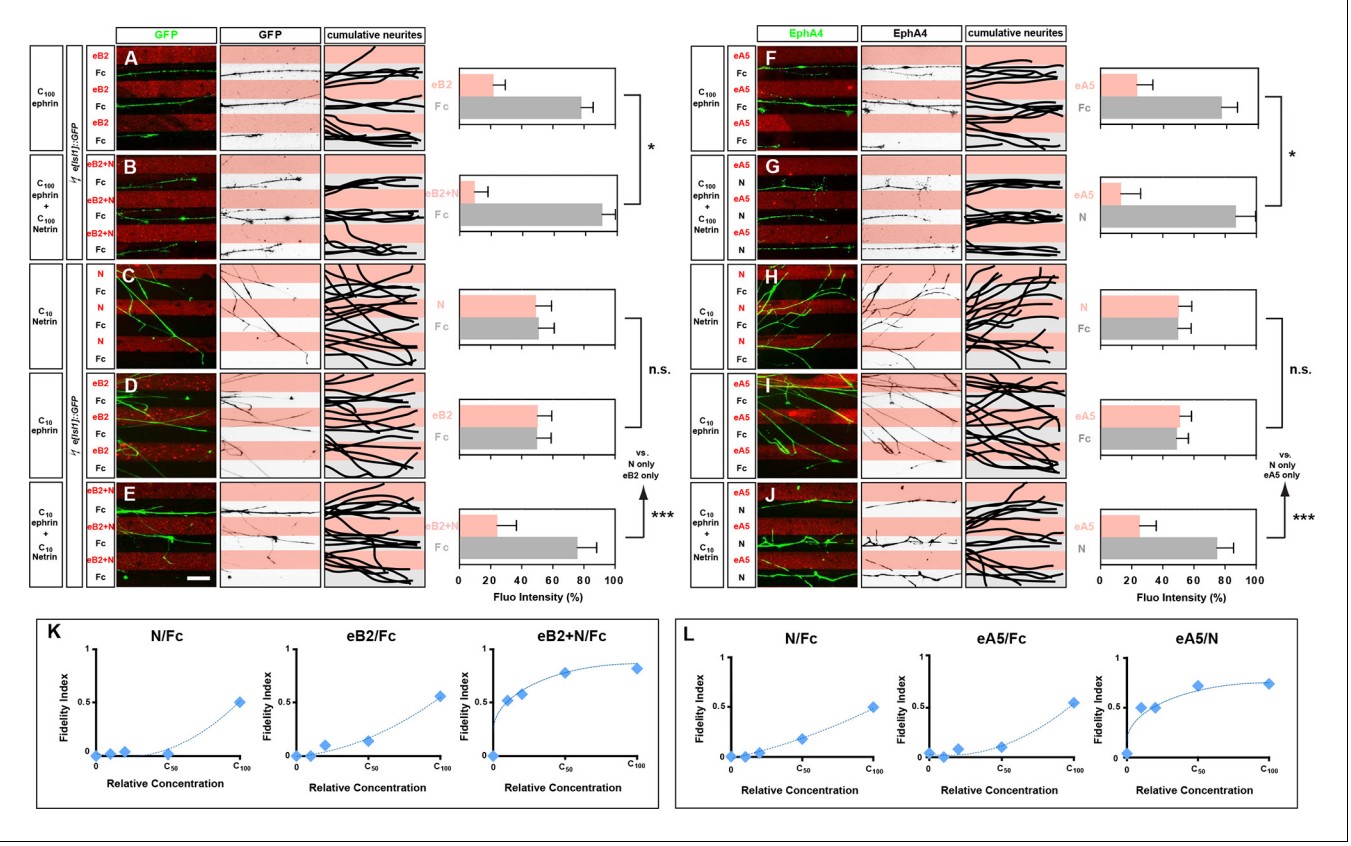

**Figure 5.** Congruent and opposing modes of Netrin and ephrin synergy in cultured LMC axons. (A–J) Left panels: explanted medial (GFP[+]) LMC neurites of *e[Isl1]::GFP*-electroporated explants on $C_{100}$ eB2-Fc/Fc (A), $C_{100}$ eB2-Fc + $C_{100}$ N/Fc (B), $C_{10}$ N/Fc (C), $C_{10}$ eB2-Fc/Fc (D), $C_{10}$ eB2-Fc + $C_{10}$ N/Fc (E) stripes and explanted lateral (EphA4[+]) LMC neurites on $C_{100}$ eA5-Fc/Fc (F), $C_{100}$ eA5-Fc/ $C_{100}$ N stripes (G), $C_{10}$ N/Fc (H), $C_{10}$ eA5-Fc/Fc (I), or $C_{10}$ eA5-Fc/$C_{10}$ N (J) stripes. Middle panels: inverted images where GFP (A–E) or EphA4 (F–J) signal is dark pixels overlaid on substrate stripes. Right panels: superimposed images of five representative explants from each experimental group highlighting the distribution of LMC neurites. Quantification of medial (GFP[+]) or lateral (EphA4[+]) LMC neurites on first (pink) and second (pale) stripes expressed as a percentage of total GFP (A–E) or EphA4 (F–J) signals. $C_{100}$ stripes were generated by incubating with ephrins at 10 µg/ml and/or Netrin-1 at 100 ng/ml. Experiments with intervening concentrations ($C_{50}$ and $C_{25}$) are shown in *Figure 5—figure supplement 1*. Minimal number of neurites: 72. Minimal number of explants: 11. (K, L) Plots of relative concentrations (x axis) over the fidelity index (y axis). Medial (K) or lateral (L) LMC neurites were challenged with one of five concentrations ($C_{100}$, $C_{50}$, $C_{25}$, $C_{10}$, 0) of ephrin or Netrin-1 to test for preferential LMC neurite growth. The fidelity index is the absolute value of: (percent growth on 2nd stripes – 50%)/50%. Index of 1 represents the complete repulsion or attraction of LMC neurites from the 1st stripes, and 0 represents no preference. Note that stripes at $C_{10}$ concentrations induced little or no preferential LMC neurite growth when only ephrin (middle plots of K and L) or Netrin-1 (left plots of K and L) was presented, but allowed a strong preferential growth of LMC neurites when both cues were present (right plots of K and L). LMC, lateral motor column; N, Netrin-1; eA5, ephrin-A5-Fc; eB2, ephrin-B2-Fc; error bars = SD; *** = p<0.001; * = p<0.05; statistical significance computed using Mann-Whitney U test; All values (mean ± SD) can be found in *Supplementary file 1B*. Scale bar: 50 µm.

The following figure supplement is available for figure 5:

**Figure supplement 1.** Netrin and ephrin synergy in LMC axon guidance.

LMC axons from Netrin-1+eB2-containing stripes towards Fc stripes, at all suboptimal concentrations studied (*Figure 5E*; 21% on eB2+Netrin-1 stripes at $C_{10}$; p<0.001 vs. Netrin-1/Fc or eB2/Fc; *Figure 5—figure supplement 1*), revealing that medial LMC axons synergistically integrate repulsion from Netrin-1 and ephrin-B2 (*Figure 5K*).

To determine the mode by which lateral LMC neurons integrate repulsive and attractive signals we studied their in vitro response to simultaneous exposure to ephrin-A5 (eA5) and Netrin-1. To do this, we quantified the preference of lateral LMC neurites for growth over Fc/eA5 stripes versus Fc stripes (eA5/Fc) or alternating stripes containing eA5 and Netrin-1 (eA5/Netrin-1), mimicking their in vivo distribution in ventral and dorsal limb mesenchyme, respectively. When challenged with eA5/Fc

stripes, lateral LMC axons preferentially grew on Fc stripes (*Figure 5F*; 77% on Fc, 23% on eA5), but when facing eA5/Netrin-1 stripes, the preference of lateral LMC axons for non-eA5 stripes was significantly increased (*Figure 5G*; 87% on Fc, 13% on eA5 + Ntn1, p<0.05 vs. eA5 only).

Next, we asked whether lateral LMC neurons integrate Netrin-1 and eA5 signaling synergistically by performing ligand titration experiments as above. Lateral LMC neurons challenged with Fc control stripes next to stripes with either Netrin-1 or eA5 at $C_{50}$ displayed a reduced preference, which was abolished at $C_{25}$, and $C_{10}$ (*Figure 5H,I*; *Figure 5—figure supplement 1*; at $C_{10}$, 52% on Netrin-1 vs. 46% on eA5; p=0.065 vs. Netrin-1/Fc or eA5/Fc). However, when Netrin-1 and eA5 were presented together in alternating stripes, each at $C_{50}$, $C_{25}$, and $C_{10}$, the repulsion of lateral LMC axons from eA5 stripes was dramatically rescued (*Figure 5J*; at $C_{10}$ 25% on eA5 and 75% on Netrin-1; p<0.001 vs. Netrin-1/Fc or eA5/Fc). These data are summarized in a fidelity index plot in *Figure 5L* and reveal that lateral LMC axons integrate attractive and repulsive responses to Ntn1 and eA5, respectively, in a synergistic manner. Together, these experiments reveal that lateral and medial LMC growth cones synergistically integrate opposing and congruent Netrin and ephrin signals.

## Ephrin-B-Netrin synergy in growth cone collapse and receptor co-localization

To begin to characterize the molecular mechanisms by which Ephrin-B2-Netrin-1 synergize, we assessed whether LMC neurons integrate Netrin-1 and ephrin signals on a short timescale, using a growth cone collapse paradigm. Explanted HH st. 24–25 chicken LMC neurons electroporated with the medial LMC marker *e[Isl1]::GFP* were exposed to Fc, eB2, Netrin-1, and eB2 and Netrin-1 for 30 min and the extent of growth cone collapse determined. A high concentration of eB2 (10 µg/ml) elicited a significant increase in collapse over controls (*Figure 6A–C*; 62% vs. 18% Fc-treated; p=0.008), and exposure to lower eB2 (1 µg/ml) or Netrin-1 (300 ng/ml) concentrations resulted in negligible collapse (p=0.194 and p=0.412, respectively, vs. Fc). Exposure of medial LMC growth cones to a mix of eB2 and Netrin-1 at low concentrations resulted in 73% collapse, significantly different from either ligand alone, or Fc controls (*Figure 6B*; p< 0.029), suggesting that the molecular mechanisms underlying ephrin-B2- Netrin-1 synergy can operate over a relatively short timescale.

The synergistic behavior of growth cones exposed to Netrin and ephrin stripes implies a crosstalk that could occur at the receptor level and/or through downstream signaling effectors. We first tested whether exposure to ephrin-B2 and/or Netrin-1 increases the levels of EphB or Unc5c receptor on the surface of LMC growth cones by treating LMC neurites with Fc, eB2, Netrin-1, and both eB2 and Netrin-1 and visualizing EphB and Unc5c receptors under permeabilizing and non-permeabilizing fixation conditions. In all cases, EphB and Unc5c expression levels remained essentially unchanged in LMC growth cones (*Figure 6—figure supplement 1*). However, immunodetection of Unc5c and EphB2 revealed a high incidence of Unc5c- and EphB-containing membrane patches, whose number and size were constant under all conditions (*Figure 6D–J*, p>0.05; *Figure 6—figure supplement 1*, data not shown [*Costes et al., 2004*]), suggesting the existence of ephrin-Netrin receptor complexes in discrete cell membrane domains of the growth cone.

## Unc5c and EphB2 form an ephrin-B2-dependent molecular complex

To test whether EphB2 and Unc5c can form a molecular complex, we performed co-immunoprecipitation experiments with lysates of HEK293 cells co-transfected with 1. *Epha3-GFP, Unc5c-Myc*; 2. *Ephb2-GFP, Unc5c-Myc*; 3. *Unc5c-Myc* fusion protein expression plasmids. In western blot analysis of cell lysates, anti-epitope tag antibodies selectively recognized their respective fusion proteins (*Figure 6—figure supplement 1L*). We found that the anti-GFP antibody precipitated Unc5c when Unc5c-Myc was co-expressed with EphB2-GFP but not with EphA3-GFP or in the absence of EphB2-GFP (*Figure 6K*). We estimated the amount of co-precipitated Unc5c-Myc to be ~15% of the amount precipitated by Myc antibodies (not shown). The Unc5c-Myc–EphB2-GFP interaction was also observed in reciprocal co-immunoprecipitation experiments (*Figure 6—figure supplement 1M*).

We next determined whether the association between Unc5c and EphB2 was ligand-dependent. Anti-GFP antibodies were used to precipitate lysates of *Ephb2-GFP* and *Unc5c-Myc* expression plasmid-transfected HEK293 cells that had been exposed to either Netrin-1, eB2-Fc, Netrin-1 and eB2-Fc, Fc or ephrin-A3-Fc (eA3-Fc). We found that the association between EphB2-GFP and Unc5c-Myc

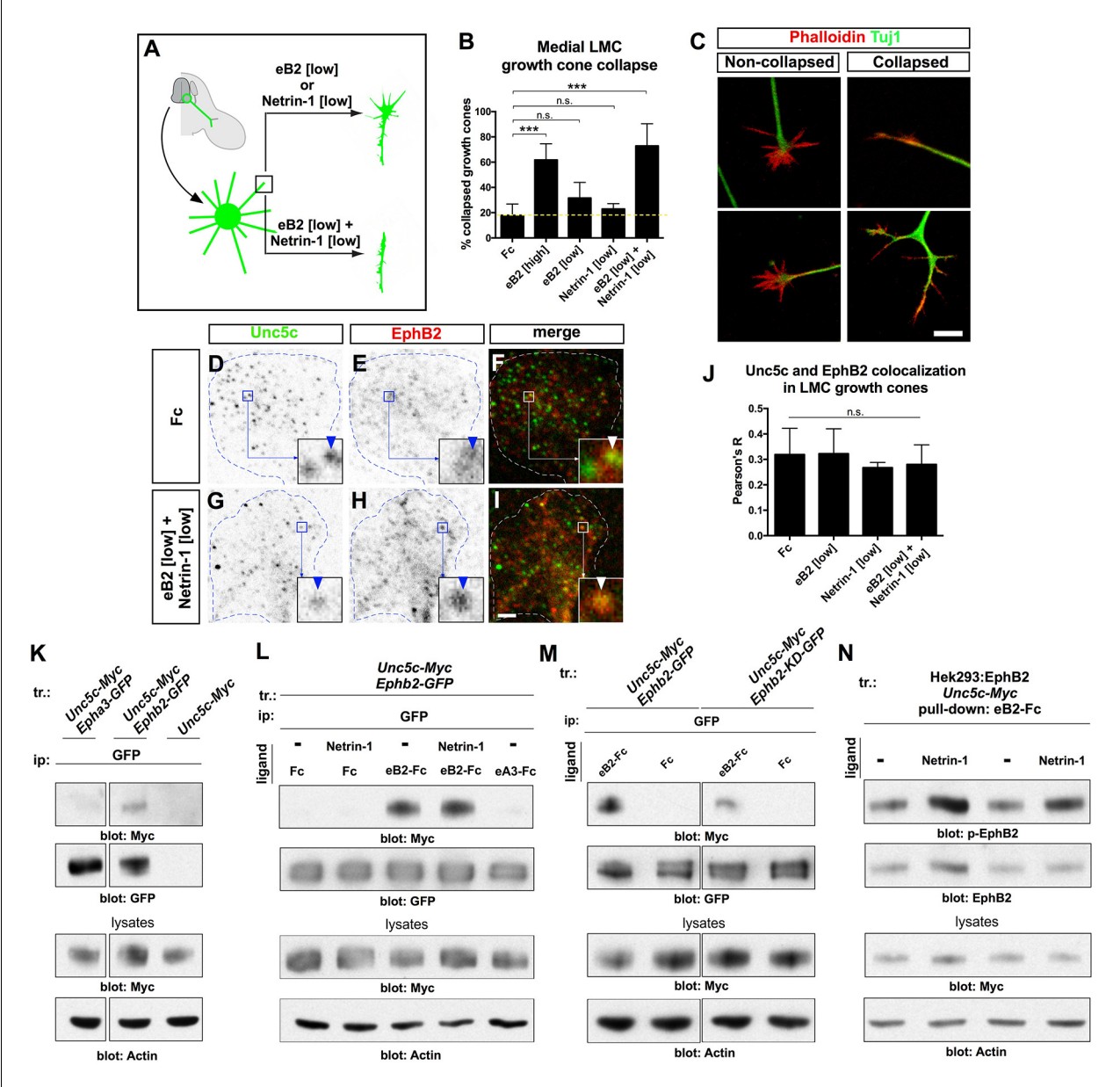

**Figure 6.** Ligand-dependent and signal-dependent EphB2-Unc5c complex formation and EphB2 phosphorylation. (**A**) Medial LMC neuron explant growth cone collapse assay scheme. (**B**) Percentage of collapsed *e[Isl1]::GFP* medial LMC growth cones following 30 min exposure to Fc (10 µg/ml), eB2-Fc (high: 10 µg/ml; low: 1 µg/ml), Netrin-1 (300 ng/ml) or Netrin-1 and eB2-Fc (300 ng/ml and 1 µg/ml). Significance computed using Fisher's exact test. (**C**) Examples of growth cones labeled with Tuj1 (green) and phalloidin (red). (**D–I**) Unc5c and EphB2 protein localization in non-permeabilized LMC growth cones treated with Fc or eB2 and Netrin-1 for 15 min. Individual channels are inverted. All treatments result in same receptor protein signal levels (eB2 and Netrin-1 images are in *Figure 6—figure supplement 1*). Receptor clusters are depicted in insets, arrowheads: Unc5c and EphB2 co-localization. (**J**) Pearson's R value as a measure of surface Unc5c and EphB2 co-localization in LMC growth cones. Co-localization levels are higher than expected by chance, as demonstrated by Costes' shuffled image P-value calculations (*Figure 6—figure supplement 1*). Ligand treatment does not increase the levels of receptor co-localization observed (p=0.7940, one-way analysis of variance (ANOVA) and Tukey's multiple comparisons test; N = 3; n ≥ 33 growth cones per treatment). (**K**) Unc5c and EphB2 receptor interactions. Unc5c-Myc was co-immunoprecipitated with EphB2-GFP but not with EphA3-GFP in transfected HEK-293 cells. All samples shown were run in same gel. (**L**) Unc5c-EphB2 interaction is selectively enhanced by 15 min incubation with eB2-Fc (1.5 µg/ml) or eB2-Fc+Netrin-1 (1.5 µg/ml +250ng/ml) but not with Netrin-1 (250 ng/ml), Fc (1.5 µg/ml), or ephrin-A3-Fc (1.5 µg/ml) prior to lysates preparation. Fc fusion proteins were pre-clustered by incubating them with anti-human or anti-mouse Ig for 1 hr at 4°C. For quantifications see *Figure 6—figure supplement 1N*. (**M**) Comparison of Unc5c interactions with wild-type or kinase-dead EphB2. Unc5c-Myc/EphB2-GFP interaction is blocked when a single point mutation is introduced in EphB2-GFP abolishing its kinase function (EphB2-KD-GFP, *Dalva et al., 2000*). For quantifications see *Figure 6—figure supplement 1O*. All samples shown were run in same gel. (**N**) P-EphB2 levels are increased upon stimulation

*Figure 6 continued on next page*

Figure 6 continued

with Netrin-1+eB2-Fc compared with eB2-Fc alone. p-EphB2 was developed first, followed by stripping of the membrane and re-blotting with anti-EphB2 antibody. Two replicate comparisons are shown; one sample t-test; p<0.02, N=10 comparisons, 4 experiments. eB2, ephrin-B2-Fc; ip, immunoprecipitation; LMC, lateral motor column; tr, transfection. All error bars = SD; *** = p<0.001; n.s. = not significant; scale bars: (C) 10 μm; (D–I) 2 μm. All values (mean ± SD) can be found in *Supplementary file 1B*.
The following figure supplement is available for figure 6:

**Figure supplement 1.** Unc5c–EphB2 receptor association.

was increased ~22 times after pre-incubation with eB2-Fc (*Figure 6L* and *Figure 6—figure supplement 1N*, p<0.01). This association was induced in a ligand-specific manner, since incubations with Netrin-1, eA3-Fc or Fc did not stimulate Netrin-ephrin receptor interactions (*Figure 6L*). Exposure to combined Netrin-1 and eB2-Fc increased EphB2-GFP–Unc5c interactions to a similar extent as eB2-Fc alone (*Figure 6L* and *Figure 6—figure supplement 1N*). Thus, a basal association between EphB2 and Unc5c is increased by ephrin-B2. Finally, since many EphB receptor functions depend on its tyrosine kinase activity (*Kullander et al., 2001*; *Soskis et al., 2012*), we addressed whether it might also be required for the Unc5c–EphB2 association. To do this, we compared the amount of Unc5c interacting with wild-type EphB2 or kinase dead point mutant EphB2 (EphB2-KD-GFP; *Dalva et al., 2000*), upon ephrin-B2 stimulation. We found that the loss of kinase function reduced the extent of interaction by ~67% when compared with the wild-type EphB2-GFP (*Figure 6M*, *Figure 6—figure supplement 1O*, p<0.01).

Eph receptor tyrosine phosphorylation induced by ephrin binding is a requirement and a correlate of their cellular activity (*Zisch et al., 1998*), leading us to examine whether following EphB2-Unc5c complex induction by ephrin-B2, the presence of Netrin-1 might potentiate EphB2 tyrosine phosphorylation. To this effect, from lysates of a cell line stably expressing EphB2 and transfected with Unc5c-Myc, we pulled down EphB2 using eB2-Fc in the presence or absence of Netrin-1 and used specific antibodies to detect tyrosine-phosphorylated EphB2 (p-EphB2; *Figure 6N*; (*Dalva et al., 2000*; *Holland et al., 1997*; *Takasu, 2002*). Exposure to both eB2 and Netrin-1 generated a significant increase of ~23% in tyrosine phosphorylation of EphB2 when compared with ephrin-B2 alone (*Figure 6N*, p<0.02). Together, our biochemical experiments indicate that concomitant stimulation with ephrin-B2 and Netrin-1 induces the formation of an EphB2-Unc5c receptor complex and increases EphB2 tyrosine phosphorylation levels, commensurate with elevated biological activity.

## Src family kinase activity is required for synergistic axon guidance responses to ephrin-B2–Netrin-1

Src family kinase (SFK) activation is critical for the intracellular relay of Eph receptor signaling, growth cone collapse, and medial LMC axon guidance in vivo (*Kao et al., 2009*; *Knoll, 2004*; *Zisch et al., 1998*). Additionally, since some studies and our experiments point to a role for SFKs in Netrin-1: Unc5 signaling (*Williams et al., 2006*), we considered SFK involvement in the integration of Netrin-ephrin activities. First, we observed that SFKs were not required for the formation of the Unc5c-EphB2 complex (data not shown). Second, we considered whether eB2 and Netrin-1 co-incidence could result in SFK activation being higher than that induced by either ligand alone. We examined the levels of SFK-activating phosphorylation (pSFK; *Boggon and Eck, 2004*), in LMC growth cones treated with Fc, eB2, Netrin-1, and both eB2 and Netrin-1. After 15 min, following ligand application but before completion of growth cone collapse, increasing doses of eB2 generated increased LMC growth cone pSFK signal that coincided with EphB receptor clusters, while Netrin-1 alone failed to induce pSFK, when compared with Fc treatment (*Figure 7—figure supplement 1*; data not shown). After 30 min of simultaneous exposure to eB2 and Netrin-1, significantly higher levels of pSFK were seen in collapsing LMC growth cones when compared with those exposed to either ligand alone or Fc (*Figure 7A–F,G*; p<0.01). Together, these experiments suggest that simultaneous exposure to eB2 and Netrin-1 might result in prolonged elevation of pSFK levels.

Finally, to assess SFK role in Netrin-ephrin responses, we performed LMC growth cone collapse and stripe assays in the presence of SFK blockers. SU6656, an SFK inhibitor, completely blocked the collapse of LMC growth cones evoked by ephrin-B2 but only attenuated by 8% the collapse caused

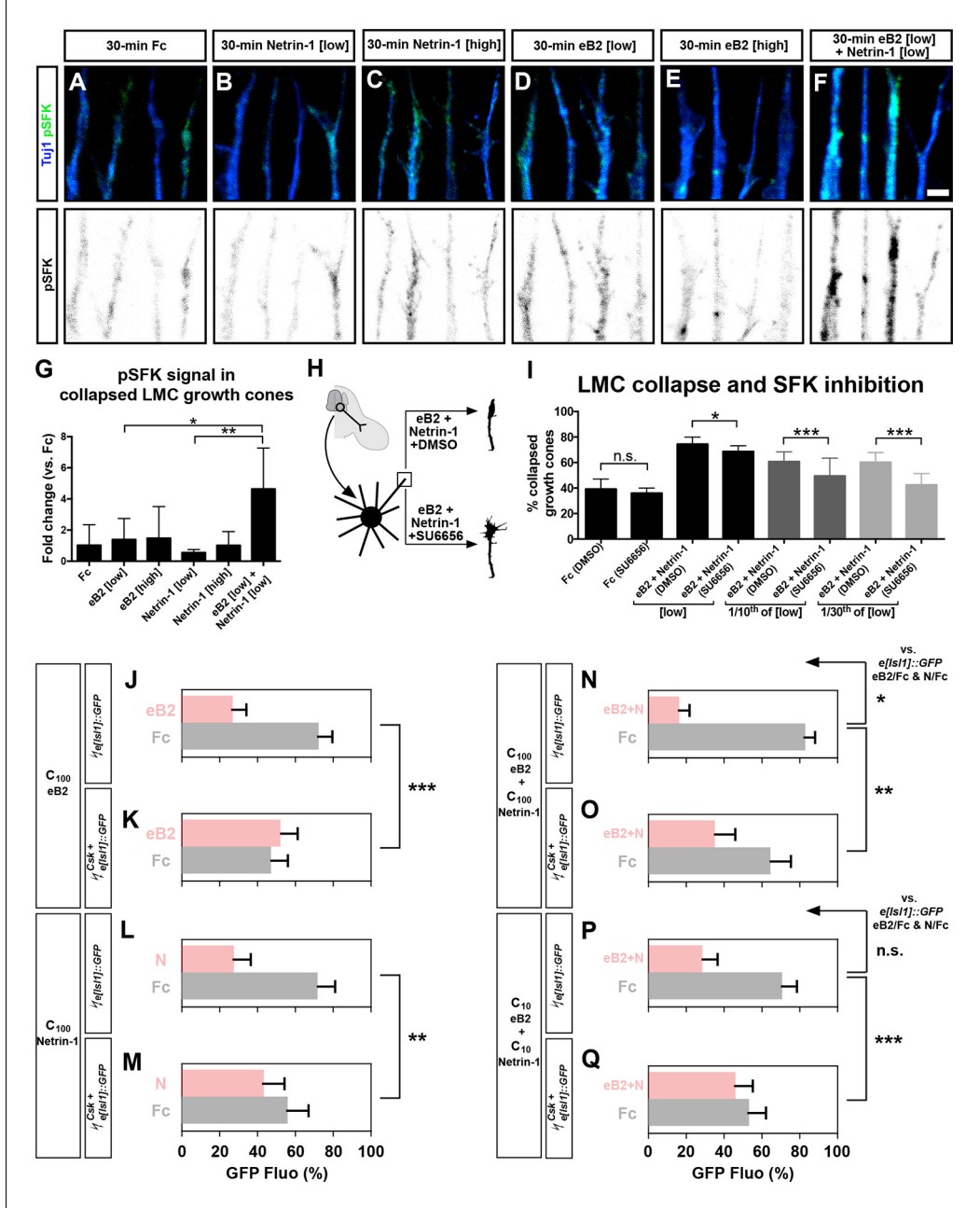

**Figure 7.** Src family kinases (SFKs) are required for synergistic repulsion from ephrin-B2 and Netrin-1. (**A–F**) Detection of pSFKs (green) and Tuj1 (blue) in collapsed growth cones after 30' treatment with Fc (10 µg/ml), low Netrin-1 (0.3 µg/ml), high Netrin-1 (1 µg/ml), low eB2 (1 µg/ml), high eB2 (10 µg/ml), or low eB2 and low Netrin-1 (1 µg/ml and 0.3 µg/ml). Bottom panels show inverted images of the pSFK channel. (**G**) Quantification of pSFK detected in collapsed growth cones treated as above. In the presence of low Netrin-1 and eB2 concentrations, pSFK signal is increased when compared with low eB2 or low Netrin-1 alone. Statistical significance computed using one-way ANOVA and Tukey's multiple comparisons test; N = 3, n ≥ 10 growth cones per condition per experiment. (**H**) LMC neuron explant growth cone collapse assay and SFK inhibition scheme. (**I**) Percentage of collapsed LMC growth cones following 30 min exposure to Fc (10 µg/ml), or Netrin-1 and eB2-Fc (0.3 µg/ml and 1 µg/ml; 1/10th of this concentration; or 1/30th of this concentration), in the presence or absence of 0.1 µM SFK inhibitor SU6656. N ≥ 3, significance computed using Fisher's exact test with n > 400 growth cones for each treatment. (**J–Q**) Quantification of medial (GFP⁺) LMC neurites of $e[Isl1]::GFP$-electroporated explants on $C_{100}$ eB2/Fc (**J**), $C_{100}$ N/Fc (**L**), $C_{100}$ eB2+N/Fc (**N**), and $C_{10}$ eB2+N/Fc (**P**), and $Csk$ and $e[Isl1]::GFP$-co-electroporated explants on $C_{100}$ eB2/Fc (**K**), $C_{100}$ N/Fc (**M**), $C_{100}$ eB2+N/Fc (**O**), and $C_{10}$ eB2+N/Fc (**Q**). Quantification of neurites

*Figure 7 continued on next page*

*Figure 7 continued*

on first (pink) and second (gray) stripes expressed as a percentage of total GFP signals. Minimal number of neurites: 80. Minimal number of explants: 12. Statistical significance computed using Mann-Whitney U test. eB2, ephrin-B2-Fc; LMC, lateral motor column; N, Netrin-1; All error bars = SD; n.s.: not significant; *: p<0.05; **: p<0.01; ***: p<0.001; scale bar: 2 μm. All values (mean ± SD) can be found in **Supplementary file 1B**.

The following figure supplement is available for figure 7:

**Figure supplement 1.** pSFK controls and Csk-electroporated axons in stripe assays.

by simultaneous eB2 and Netrin-1 application (*Figure 7H–I*; p<0.05; *Figure 7—figure supplement 1*; data not shown; *Blake et al., 2000*). The combination of SU6656 with eB2 and Netrin-1 at 1/10th and 1/30th of the above concentrations, resulted in a 19% and 29% attenuation of synergistic collapse, respectively (*Figure 7I*, p<0.001). In stripe assays, medial LMC neuron over-expression of Csk, an inhibitory kinase of Src (*Imamoto and Soriano, 1993*), completely blocked responses to Fc/N and Fc/eB2 stripes at $C_{100}$ (*Figure 7J–M*, p<0.01), while the preference elicited by Fc/N+eB2 stripes at $C_{100}$ concentration was attenuated by 57% (*Figure 7N–O*, p<0.01) and it was reduced by 87% when challenged with Fc/N+eB2 stripes at $C_{10}$ (*Figure 7P–Q*; p<0.001; detailed quantification: *Supplementary file 1B*). Together, these results indicate that SFK activation is a key step in the integration of ephrin-B2 and Netrin-1 signaling that leads to elevated repulsion of LMC growth cones.

## Discussion

Our genetic in vivo and in vitro assays argue that 1. acting through its attractive and repulsive receptors, Netrin-1 guides spinal motor axons in the limb, 2. Netrin-1 and A-class and B-class ephrins synergize in attractive-repulsive (opposing) and repulsive-repulsive (congruent) modes to guide motor axons, 3. Unc5c and EphB receptors form an ephrin-B ligand-dependent and Eph tyrosine kinase-dependent complex, and 4. congruent synergy involves the potentiation of common effectors of ephrin and Netrin signaling pathways such as SFKs. Here, we discuss these findings in the context of the logic of motor axon guidance, Netrin receptor function, and axon guidance signal integration.

### Netrin-1, synergy and other motor axon guidance signals

Our results argue for a model in which Netrin-1 present in the dorsal limb mesenchyme concomitantly directs lateral LMC axons into dorsal limb nerves, and medial LMC axons into ventral limb nerves. Our most direct evidence of this is the contrasting response of cultured medial and lateral LMC axons to Netrin-1 protein, which depends on (1) the selective expression of Unc5c in medial LMC neurons, and (2) the expression of attractive Netrin receptors in LMC neurons. Unc5c loss results in decreased repulsion from Netrin-1 in vitro and entry of medial LMC axons into the dorsal limb in vivo while inhibition of attractive Netrin receptors leads to a loss of lateral LMC axon preference for growth on Netrin-1. In line with previous data (*Hong et al., 1999*), Unc5c signaling dominates over attraction to Netrin-1, since over-expression of Unc5c in LMC neurons results in more LMC axons entering the ventral limb. Following this logic, the growth of *Unc5c* mutant medial LMC axons into the dorsal limb of mice might be caused by the loss of the dominant repulsive receptor uncovering Netrin attraction. However, since in vitro loss of Unc5c function in LMC neurons does not uncover an increased preference for Netrin-1 stripes, it is plausible that attractive Netrin-1 receptors in medial LMC axons are not functional in the context of our assays.

Following dorsoventral limb trajectory selection, LMC axons are constrained to the permissive axonal pathways in the limb where they make subsequent trajectory selections that bring them to their muscle target (*Tosney and Landmesser, 1985*). During these stages, Netrin receptor expression in LMC neurons is dynamic and Netrin-1 is expressed in the distal limb mesenchyme, suggesting that it is involved in the selection of muscle nerve trajectory by LMC axons as implied by muscle nerve trajectory defects in Unc5c mutants (S.P. and Thomas Jessell; unpublished observation). Thus, along with its regulation of motor axon exit from the nervous system (*Bai et al., 2011*; *Varela-Echavarria et al., 1997*), and in contrast to other signals that apparently guide subpopulations of LMC axons (*Chai et al., 2014*; *Huber et al., 2005*; *Kramer et al., 2006*), Netrin-1 appears to be a

pervasive signal controlling motor axon guidance in vertebrates and non-vertebrates (*Winberg et al., 1998*).

## LMC axon guidance and attractive Netrin-1 receptors

Limb rotation experiments predict that a localized axon guidance signal constrains LMC axon trajectory choice at the base of the limb (*Ferns and Hollyday, 1993*), thus excluding Netrin-1's long-range diffusible axon guidance activity as a dorsoventral nerve selection signal. However, Netrin-1 can also act as a short range, non-diffusible signal (*Brankatschk and Dickson, 2006*; *Timofeev et al., 2012*) and in our in vitro assays, medial and lateral LMC axons exhibit robust and specific responses to immobilized Netrin-1. Additionally, since Netrin-1 protein localization in the limb matches closely that of its mRNA, Netrin-1 likely functions as a contact-dependent axon guidance cue for LMC growth cones at the limb nerve bifurcation abutting the Netrin-1 expression domain. One corollary of this, based on non-vertebrate models of Netrin repulsion (*Keleman and Dickson, 2001*), would be the lack of requirement for attractive Netrin receptors for short range Unc5c-mediated LMC axon guidance, in agreement with our genetic experiments showing that DCC is dispensable for Unc5c-mediated medial LMC guidance. *Unc5c* mutation causes more severe medial LMC guidance defects than *Ntn1* mutation. One possible explanation is that other Unc5 ligands such as FLRTs might contribute to Unc5c repulsion of LMC axons (*Yamagishi et al., 2011*). In addition, the hypomorphic nature of the *Ntn1*[Gt] mutation (*Serafini et al., 1996*), with residual Netrin-1 protein guiding some medial LMC axons in *Ntn1* mutants, could account for the weaker misprojection phenotype.

Neogenin has been long proposed to be a Netrin-1 receptor; however, most direct functional evidence of its function in axon guidance implicates it as a repulsive guidance molecule (RGM) receptor (*Rajagopalan et al., 2004*; *Xu et al., 2014*). Our in vitro antibody blocking experiments directly demonstrate that Neo1 function is required for Netrin-1-mediated axon guidance and the functional equivalence of DCC and Neo1. The rescue of Neo1 antibody block by DCC expression also implies that the intracellular effectors of Netrin-1:DCC attraction signaling are present in chicken lateral LMC neurons.

Contrary to our in vitro experiments where Neo1 loss of function results in loss of growth preference on Netrin-1, genetic loss of *DCC* and nearly all of *Neo1*, or *Dscam*, does not lead to any lateral LMC guidance defects, suggesting that in vivo, attractive Netrin-1 receptors are dispensable for the fidelity of lateral LMC axon guidance. How can the in vitro and in vivo data be reconciled? One possibility is that non-Netrin-1 lateral LMC guidance cues such as ephrin-As in the ventral limb are operational even when Netrin-1 attractive receptors are removed, preserving the high fidelity of lateral LMC axon trajectory choice. It is also possible that the 10% of wild-type *Neo1* protein levels produced from the *Neo1* hypomorphic allele (*Bae et al., 2009*) are sufficient to elicit normal attraction to Netrin-1 in lateral LMC axons and synergize with other axon guidance cues in the limb.

## Mechanisms of ephrin-Netrin synergy

Our present experiments show that coincidence of Netrin-1 and ephrin is integrated in a synergistic manner by spinal motor neurons leading to robust preference responses. Our argument against simple additivity of ephrin and Netrin-1-evoked responses is based on the quantitative analysis of neurite growth preferences. Thus, observable effects from the combination of ligands, each at concentrations of less than half of a sub-threshold concentration (without effect on its own), implies synergy and not additive mechanisms. Our stripe assays exhibited similar overall outgrowth in all conditions, and detected a robust axon guidance response to ephrin and Netrin-1 even at 1/10th of concentrations insufficient for ephrin or Netrin-1 to elicit an effect on their own. Moreover, these responses reflect the bifunctional nature of Netrin-1's chemotropic activity: Netrin-1 attraction synergizes with ephrin-A repulsion of lateral LMC neurons while Netrin-1 repulsion synergizes with ephrin-B repulsion of medial LMC axons. Thus, contrasting modes of synergistic integration occur in related populations of spinal motor neurons, possibly reflecting the evolutionary advantage of axon guidance synergy. Furthermore, it is likely that the molecular mechanisms that integrate two repulsive cues, and an attractive cue with a repulsive cue are fundamentally different, even if the cues are molecularly related as is the case for ephrin-As and ephrin-Bs.

Synergy implies a cross-talk between Netrin and ephrin signaling, which could be initiated at the receptor level or at downstream signaling nodes. Our experiments suggest a two-step model of

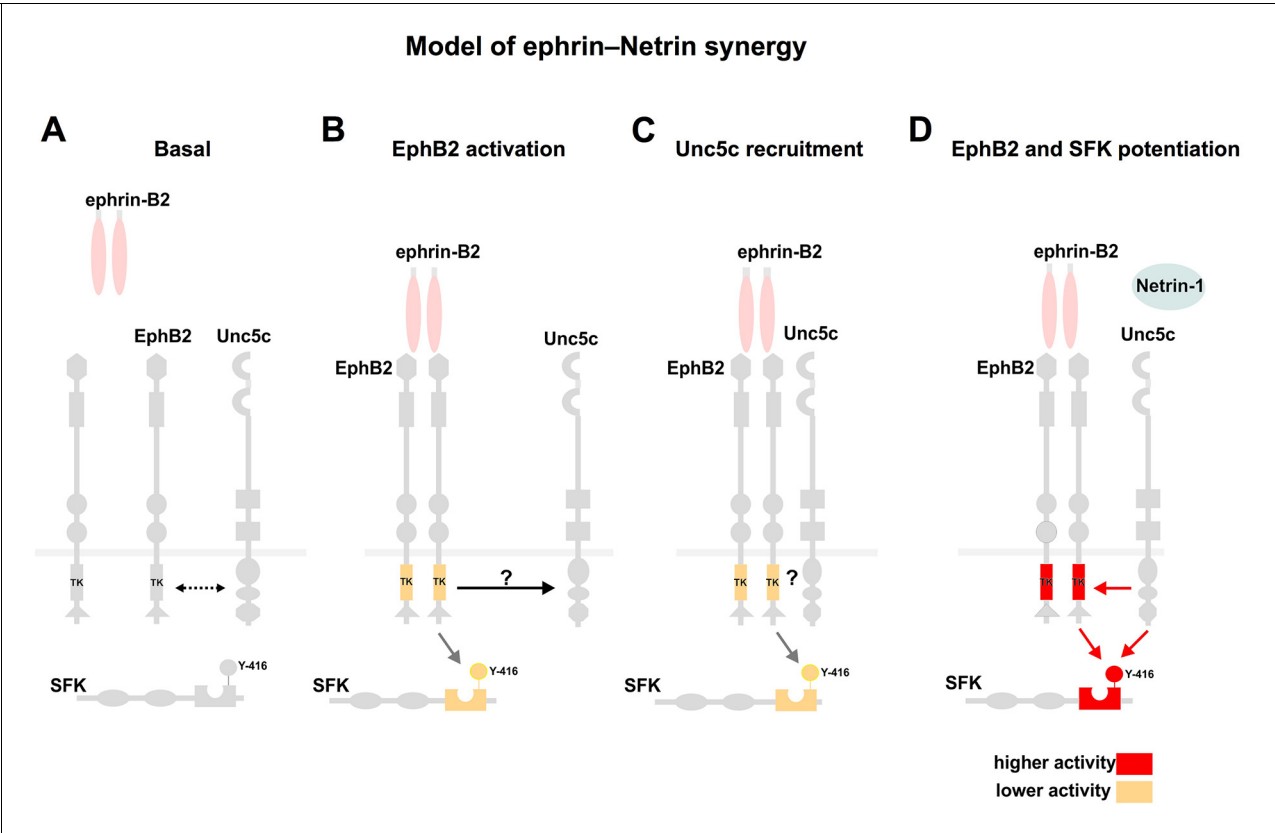

**Figure 8.** Model summarizing EphB2 interactions with Unc5c. (A) Under non-stimulated conditions there is a low level interaction between EphB2 and Unc5c (depicted by dotted two-directional arrow). (B, C) Upon ephrin-B2 stimulation, signaling through EphB2 kinase activity induces direct or indirect association (arrow) with Unc5c. (D) Netrin-1 signals through the novel receptor complex resulting in elevated EphB2 phosphorylation and, together with EphB2, in SFK potentiation. SFK, Src family kinase; TK, tyrosine kinase domain. Y-416, tyrosine-416 of Src, whose phosphorylation positively correlates with Src kinase activity.

congruent ephrin-Netrin synergy, likely starting at the receptor level (*Figure 8*). As a first step of our model, independent of ligands, a considerable fraction of EphB and Unc5c receptors on the surface of medial LMC growth cones is apparently found in the same membrane compartment, contrasting the non-overlapping distribution of some ephrins and Eph receptors (*Marquardt et al., 2005*). Such membrane compartment co-existence might facilitate the formation of EphB2-Unc5c complexes following the addition of ephrin-B2, as detected in our biochemical assays. Importantly, this effect is ligand-specific, and signaling through the EphB2 receptor is important since tyrosine kinase point mutation inhibits complex formation. The second step involves the action of Ntn1 alone or in conjunction with ephrin-B2, through EphB2-Unc5c complexes, resulting perhaps in the activation of novel downstream signaling pathways, or increased activation of common intracellular effectors. In support of the second possibility, Netrin-1 presence increased EphB2 tyrosine-phosphorylation, a correlate of Eph receptor activation required for many of its functions (*Zisch et al., 1998*). We also found that the coincidence of ephrin-B2 and Netrin-1 resulted in prolonged activating phosphorylation of SFKs when compared with that induced by ephrin-B2 alone, and that SFK function is required in LMC neurons for ephrin-B2-Ntn1 synergistic responses. Thus, since *Src* mutation abolishes the fidelity of medial LMC limb nerve selection (*Kao et al., 2009*; *Liu et al., 2004*), the second step might entail increased EphB2 and Src kinase activity leading to more intense activation of common signaling pathways. The fact that SFK inhibitors could not totally block repulsive growth cone responses upon Netrin-1-ephrin-B2 stimulation raises the interesting possibility that additional, non-SFK mechanisms integrating Netrin-ephrin signaling exist. Furthermore, the genetic observation that in EphB1 and EphB3 mutants many medial LMC axons can be labeled from dorsal limb muscles

(*Luria et al., 2008*), raises the question of whether the EphB2-Unc5c interaction is a general property of B-class Eph receptors.

Our experiments also shed light on the significance of synergistic axon guidance. With additive signals, blocking one completely would have only partial effect on the overall fidelity of LMC trajectory selection. However, in the case of synergistic signals, taking out most of a signal might still leave enough of it to synergize with its partner pathway since this effect can occur at suboptimal signal levels. Completely inactivating a synergizing signal will lead to extreme effects where its partner might not function efficiently enough to maintain LMC axon pathway selection fidelity. In this view limb-expressed GDNF and ephrin-A appear as additive signals, such that deleting one of them results only in a partial loss of LMC trajectory choice fidelity (*Kramer et al., 2006*). However, genetic perturbation of ephrin and Netrin signaling in medial LMC axons suggest a synergistic interaction: loss of EphB signaling leads to severe medial LMC axon guidance phenotypes (*Luria et al., 2008*), as does the loss of Unc5c in the present study, arguing against redundancy between the two systems, and together with our in vitro data, for synergy. Synergistic integration of axon guidance cues might endow neurons with a powerful means of preventing potential axon guidance errors caused by fluctuations in expression levels of guidance receptors and their ligands.

## Experimental procedures

### Animals
Mouse lines were as previously described: $Unc5c^{rcmTg(Ucp)1.23Kz}$ (*Ackerman et al., 1997*; *Burgess et al., 2006*); $Dcc^{tm1WBG}$ (*Fazeli et al., 1997*); $Ntn1^{Gt(pGT.8TM)629Wcs}$ (*Serafini et al., 1996*); $Neo1^{Gt}$ (*Bae et al., 2009*); *Unc5a* (*Williams et al., 2006*); *Dscam* (*Amano et al., 2009*). Timed mating vaginal plug was designated as e0.5. Fertilized chicken eggs (Couvoir Simetin) were incubated at 38°C and staged according to *Hamburger and Hamilton, 1951*.

### Chick in ovo electroporation
Chick spinal cord electroporation of expression plasmids or siRNAs was performed at HH st. 18/19 as described (*Croteau and Kania, 2011*). siRNA sequences used (sense strand): [*Unc5c]siRNA*, 1:1:1 mixture of GAACCCGAAGAAGCTTACATCGTGA, CAGCAACTGTGATTGTCTATGTGAA, and CACCGTGACTTTGAGTCAGATATTA. *Dcc* and $Dcc^{ΔICD}$ expression plasmids were described previously (*Hong et al., 1999*) and were co-electroporated with the *GFP* expression plasmid at a 1:10 molar ratio.

### HRP retrograde tracing of motor neurons
Retrograde labeling of mouse or chick motor neurons using HRP (Roche) or conjugated dextran (Invitrogen) as tracers was performed as described (*Luria et al., 2008*).

### In vitro stripe assay
Protein carpets were produced using silicon matrices (*Knöll et al., 2007*). Carpets contained an alternating stripe pattern of ephrin-Fc, Netrin-1, or Fc only as controls: the first stripe was labeled with Fc-specific Cy3-conjugated antibody (4:1 weight ratio) while the second stripe contained unconjugated Fc-specific antibody. Dissection of E5 chick spinal motor column and dissociated culture was as described (*Kao and Kania, 2011*).

### In situ mRNA detection and immunostaining
In situ mRNA detection, immunofluorescence and live-cell staining were performed as described (*Kao and Kania, 2011*) or using standard methods. Probes are available upon request. A rabbit polyclonal anti-Unc5c antiserum was raised against a c-terminal peptide of mouse Unc5c: CGRHETVVSLAAEGQY. The antiphospho-Tyr$^{416}$-Src antiserum (Life Technologies) also reacts with Yes and Fyn. See *Supplementary file 1A* for all antisera.

For non-permeabilized assays, tissue was exposed to ligands for 15 min and then placed on ice, and a 5-min pre-blocking step was performed by replacing half the media with phosphate-buffered saline (PBS) containing 2% bovine serum albumin (BSA) (final, 1% on tissue) and incubating at 4°C. Half of the media was then replaced with motor neuron media containing primary antibodies

against Unc5c (final dilution of 1 in 1000), EphB2 (1 in 1000), and EEA1 (1 in 500) as control (fluorescent-conjugated phalloidin (1 in 400) was also used as control in some experiments due to species cross-reaction issues with the EEA1 antibody) and tissue was incubated 30 min at 4°C. Tissue was then fixed with a mixture of $\frac{1}{5}$th 30% sucrose and $\frac{4}{5}$ths 4% PFA for 15 min at 4°C. Three quick washes with PBS were followed by replacing half the PBS with PBS containing secondary antibodies (final, 1 in 1000) and a 1-hr incubation at 4°C. Finally, three quick washes were followed by mounting in Mowiol. For the permeabilized control, fixation occurred after ligand incubation and before primary antibody staining, primaries were added in motor neuron media with added triton (0.3%), and secondaries in PBS with added triton (0.3%). Otherwise, all concentrations, incubation times, and temperatures were identical.

### Image quantification and statistical analysis

GFP and PLAP-labeled axonal projection, protein, and mRNA expressions, and motor neuron numbers of limb section images were quantified using Photoshop (Adobe) or ImageJ (NIH) as described (*Kao et al., 2009*). Stripe assays were quantified as previously described (*Kao and Kania, 2011*). Data from the experimental replicates were evaluated using Microsoft Excel. Experimental measurements were compared using statistical tests as indicated in figure legends, with 0.05 a significance threshold.

### Co-immunoprecipitation and western blotting

HEK293 cells (ATCC CRL 1573) were transfected using the calcium phosphate method, combining the following plasmids as described in results: *Ephb2-GFP* (*Kao et al., 2009*), *mUnc5c-Myc* (gift of Franck Polleux), *mDcc-HA* (*Stein, 2001*), *Epha3-GFP* (Gift of Dimitar Nikolov) and *Ephb2-KD-GFP* (*Dalva et al., 2000*). In experiments with ligand incubations, after ~40 hr the media was replaced and cells were starved for 2 hr in OPTIMEM (Gibco), followed by incubation in Fc (1.5 µg/ml; R&D), Netrin-1 (250 ng/ml, R&D), eB2 (1.5 µg/ml, R&D), eA3 (1.5 µg/ml, R&D) or ligand combinations for 15 min at 37°C. Fc and Fc-fusion ligands were pre-clustered by mixing them with anti-Fc antibody (human for Fc and eB2 and mouse for eA3) for 1hr at 4°C. After a wash in PBS, cell lysates were prepared in lysis buffer containing 10mM Tris-Cl pH 8, 137 mM NaCl, 2mM EDTA, 1% NP40 and proteinase inhibitors (cOmplete ULTRA Tablets, Mini, EDTA-free, EASYpack, Roche). Immunoprecipitations were performed overnight by binding pre-cleared lysates (centrifuged two times at 14,000 rpm for 15 min and 5 min, respectively) to ProteinA/G agarose that were previously incubated for 2 hr at 4°C with the corresponding antibody. After two washes in 0.1% NP40 in PBS and one in PBS, samples were separated on 4–12% gradient Bis-Tris polyacrylamide gels (Nupage Novex, Life Technologies). Western blots were incubated overnight with the indicated antibodies diluted in 3% milk and developed with ECL reagent (Pierce). For detection of p-EphB2, *Ephb2*-stably transfected HEK293 cells (*Poliakov et al., 2008*; gift from Tony Pawson, cell line identity not authenticated) were transfected with Unc5c-Myc and starved overnight in OPTIMEM (Gibco) prior to treatment with eB2-Fc (0.15 µg/ml) or eB2-Fc and Netrin-1 (0.15 µg/ml + 0.6 µg/ml; [*Holland et al., 1997*]]) for 15 min. Cell lysates were prepared in buffer, pre-cleared as described and bound overnight to ProteinA/G agarose. Samples were run in 4–12% gradient Bis-Tris polyacrylamide gels. Detection of p-EphB2 (*Dalva et al., 2000*; *Holland et al., 1997*) was followed by membrane stripping (Thermo Scientific) for 20 min at 37°C, blocked in 3% BSA and re-blotted with a goat anti-EphB2 antiserum (R&D systems). Pixel intensity and area of western blot unsaturated bands were measured in Photoshop and total intensity calculated. P-EphB2 signals were normalized to EphB2 and Netrin-1+ eB2-Fc vs. eB2-Fc ratios calculated. In the case of immunoprecipitation experiments, the amount of Unc5c was normalized to immunoprecipitated EphB2-GFP. Statistics were performed with a one-sample t-test, with the null hypothesis mean=1.

## Acknowledgements

The authors thank T. Jessell for his generosity and discussions and S. Butler for bringing our attention to Netrin-1 expression in the limb. The authors also thank T. Jessell, S. Butler, M. Cayouette, B. Novitch, A. Tufford and C. Law for critical comments on the manuscript. J. Cardin, Q. Liu and M. Liang for technical assistance and L. Delorme for secretarial assistance. *Dscam* null mice were a kind gift of K. Yamakawa. S.P. was a Helen Hay Whitney Fellow in the laboratory of T. Jessell supported

by the Howard Hughes Medical Institute, and grants from the NINDS RO1 NS033245, ProjectALS/ P2ALS and The G Harold & Leia Y Mathers Charitable Foundation. L-P.C. was a recipient of an FRSQ Bourse de Formation and D.M. was a recipient of a Mexican Council for Science and Technology (CONACYT) graduate scholarship. S.L.A. is a Howard Hughes Medical Institute Investigator. This work was supported by a grant from the Canadian Institutes of Health Research (MOP-97758 and MOP-77556), the EJLB foundation, Brain Canada, Canadian Foundation for Innovation, and the W. Garfield Weston Foundation to A.K. who was also an FRSQ Chercheur-boursier Senior, and by a grant from the Ministry of Science and Technology, R.O.C. (MOST) (104-2311-B-038 -004) to T-J.K.

## Additional information

### Funding

| Funder | Grant reference number | Author |
|---|---|---|
| Canadian Institutes of Health Research | MOP-97756 | Artur Kania |
| Canadian Institutes of Health Research | MOP-77556 | Artur Kania |
| Helen Hay Whitney Foundation | | Sebastian Poliak |
| Canadian Institutes of Health Research | | Jean-François Cloutier Frederic Charron |
| Fonds de Recherche du Québec - Santé | | Louis-Philippe Croteau |
| EJLB Foundation | | Artur Kania |
| Mexican Council for Science and Technology | | Daniel Morales |
| Howard Hughes Medical Institute | | Susan Morton Susan L Ackerman |
| Ministry of Science and Technology, Taiwan | 104-2311-B-038 -004 | Tzu-Jen Kao |

The funders had no role in study design, data collection and interpretation, or the decision to submit the work for publication.

### Author contributions

SP, DK, T-JK, AK, Conception and design, Acquisition of data, Analysis and interpretation of data, Drafting or revising the article; DM, L-PC, Acquisition of data, Analysis and interpretation of data, Drafting or revising the article; EP, Conception and design, Acquisition of data, Analysis and interpretation of data; SM, Acquisition of data, Analysis and interpretation of data, Contributed unpublished essential data or reagents; J-FC, Acquisition of data, Analysis and interpretation of data, Drafting or revising the article, Contributed unpublished essential data or reagents; FC, MBD, SLA, Analysis and interpretation of data, Drafting or revising the article, Contributed unpublished essential data or reagents

### Ethics

Animal experimentation: All experimental procedures were approved by the Animal Care Committee at the Institut de Recherches Cliniques de Montréal, in accordance with the regulations of the Canadian Council on Animal Care. The following protocol reference numbers were used: 2005-03, 2008-18, 2009-10, 2011-30 and 2012-22.

## Additional files

**Supplementary files**

• Supplementary file 1. Antisera, recombinant proteins, and quantification values. (A) Table listing all antibodies and recombinant proteins used. (B) Values of quantifications in main figures. (C) Values of quantifications in figure supplements.

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
