## [Decision Letter]

Thank you for submitting your work entitled "Synergistic integration of Netrin and ephrin axon guidance signals by spinal motor neurons" for consideration by *eLife*. Your article has been reviewed by two peer reviewers, one of whom is a member of our Board of Reviewing Editors, and the evaluation has been overseen by the Reviewing Editor and a Senior Editor.

The reviewers have discussed the reviews with one another and the Reviewing editor has drafted this decision to help you prepare a revised submission.

Summary:

In this study Poliak et al. demonstrate a requirement for Unc5c repulsive signaling in response to Netrin-1 during LMC motor axon guidance. Through a series of in vitro and in vivo experiments, they propose that Unc5c-mediated repulsion has a synergistic effect when combined with EphB2 signaling, and that this may in part be due to convergence on Src kinase activation. The analysis of the in vivo phenotypes is clear and compelling, and increases our understanding of the role of Netrin receptors during motor axon guidance. The in vitro stripe assay and growth cone collapse assay are powerful and allow for precisely measuring the effect of combining different cues on axon behavior. How a growth cone integrates its responses to different extracellular cues remains largely unknown. Both in vivo and in vitro data support the hypothesis that synergy occurs between Unc5c and EphB2 signaling in LMC motor neurons, and although a mechanistic understanding of how this synergy is achieved remains incomplete, this work addresses an important question and lays the foundation for future research investigating these mechanisms in more detail.

In mice and chick embryos, Netrin-1 is enriched in the dorsal limb and Unc5c is expressed in ventrally projecting LMC-m neurons at stages when LMC axons select their trajectory. While all LMC neurons express DCC or Neogenin, there are no guidance defects in the dorsally projecting LMC-l axons in the absence of Netrin or its attractive receptors. In contrast, a significant number of LMC-m axons mis-project in the absence of Netrin or Unc5c (though DCC is not required for LMC-m guidance). In vitro, chick LMC-m axons are repelled by Netrin1+ stripes in an Unc5c-dependent manner, whereas LMC-l axons are attracted to Netrin1+ stripes in a Neogenin-dependent manner. Combining low doses of ephrinB2 and Netrin1 (less than half of the sub-threshold concentrations of each) results in a significant avoidance response by LMC-m axons; a similar effect is observed on growth cone collapse. Similarly, LMC-l axons are more strongly attracted to sub-threshold amounts of Netrin-1 when they are alternated with sub-threshold amounts of ephrinA5. Though the mechanism for this synergistic effect is not further explored, the authors undertake several experiments addressing the relationship between Unc5c and EphB2. Unc5c and EphB2 form a complex when over-expressed in vitro, and this interaction is increased upon ephrinB2 treatment and requires EphB2's kinase activity. Netrin treatment does not enhance the interaction between Unc5c and EphB2, but increases the phosphorylation levels of EphB2. Finally, the authors propose that signaling downstream of both receptors may converge on Src kinases, as treatment of LMC-m axons with both ligands for a sustained period of time increases the levels of phospho-Src compared to treatment with either alone. Growth cone collapse in response to ephrinB2 and Netrin1 is partially suppressed by treating with a Src kinase inhibitor, and over-expression of Csk suppresses LMC-m avoidance of Netrin1+ and ephrinB2+ stripes, providing important functional data to support the hypothesis that both pathways act through Srcs.

Essential Revisions:

1) The authors suggest that the relatively weak dorsal limb mis-projection phenotype (relative to Unc-5) likely results from residual Netrin expression in the Netrin gene trap allele. It would be nice if it were possible to analyze these phenotypes in the recently published Netrin null mice described by Kennedy, Kania and colleagues. It is acknowledged that this may not be possible, however, depending upon the viability of the null mutation.

2) Kao et al. 2009 previously showed that the EphB2 gain of function phenotype could be suppressed by co-expression of the Src inhibitor Csk. Performing a similar experiment with Unc5c gain of function would complement the functional experiments demonstrating a role for Src kinases downstream of ephrinB and Netrin in vitro, and greatly strengthen the model that EphB and Unc5 converge on Src activation in vivo.

---

## [Author Response]

*1) The authors suggest that the relatively weak dorsal limb mis-projection phenotype (relative to Unc-5) likely results from residual Netrin expression in the Netrin gene trap allele. It would be nice if it were possible to analyze these phenotypes in the recently published Netrin null mice described by Kennedy, Kania and colleagues. It is acknowledged that this may not be possible, however, depending upon the viability of the null mutation.*

This comment refers to the study by Bin, J. M. et al. Complete Loss of NetrinJ1 Results in Embryonic Lethality and Severe Axon Guidance Defects without Increased Neural Cell Death. Cell Rep 12, 1099–1106 (2015). Unfortunately, most *Netrin1* nulls die around e10.5/e11.5 while some escapers make it to e14.5. In order to compare the incidence of LMC projection errors in *Netrin1* null mice to those reported in the present manuscript, we envisioned doing retrograde labelling from limb muscles. This analysis is normally done at e12.5 (forelimb) or e13.5 (hindlimb) and thus would take more than six months to complete given mutant mice availability and the rare occurrence of *Netrin1* null embryos of those ages. Crosses of the *Netrin1* nulls into different genetic backgrounds are ongoing. Alternatively, we could cross the ZJ Crest::alkaline phosphatase medial LMC reporter into the *Netrin1* null background and study e11.5 embryos, but this analysis would not quantify lateral LMC errors, and would need two mouse generations.

While doing the analysis for Bin et al., we were able to obtain some e11.5 limb tissue which we stained for Unc5c in order to track medial LMC axons. In the few sections that contained dorsal and ventral limb nerves, we quantified the incidence of *Unc5c^+^* axons in the dorsal limb nerve as a proportion of total Unc5c signal in the dorsal and ventral limb nerves. These data are appended to the end of this letter, and indicate that more *Unc5c^+^* axons enter the dorsal limb of *Netrin1* nulls than those of controls. However, this experiment is difficult to interpret since Netrin1 absence could be locally affecting Unc5c expression levels, as it does for *Dcc* (Bin et al.), and relatively few limb sections were available for quantification. Also, in contrast to retrograde labelling, this experiment does not allow us to make a quantitative comparison with the *Unc5c* mutant phenotype. Finally, as we emphasized in the Discussion, the reduced phenotype in **Ntn1*^Gt^* mutants (compared to *Unc5c* mutants) could be due to the presence of redundant Unc5c ligands in the dorsal limb. To be fairer, we now give equal importance to both possibilities in the Discussion.

2) Kao et al. 2009 previously showed that the EphB2 gain of function phenotype could be suppressed by co-expression of the Src inhibitor Csk. Performing a similar experiment with Unc5c gain of function would complement the functional experiments demonstrating a role for Src kinases downstream of ephrinB and Netrin in vitro, and greatly strengthen the model that EphB and Unc5 converge on Src activation in vivo.

We did this experiment and it is now incorporated in the manuscript as part of Figure 3, Results section (paragraph three, subsection “Medial and lateral LMC axons synergistically integrate congruent and opposing Netrin and ephrin signals”) and [Supplementary-material SD1-data]. Overexpression of Unc5c in lateral LMC axons resulted in their repulsion from Netrin stripes. We were able to block this effect by overexpressing Csk in LMC neurons. We are grateful to the reviewers for this observation that strengthens our argument about signal convergence at Src.